# Training Transitive and Commutative Multimodal Transformers with LoReTTa

**Manuel Tran**[1,3,4]    **Yashin Dicente Cid**[2]    **Amal Lahiani**[1]    **Fabian J. Theis**[3,4]

**Tingying Peng**[4,*]                    **Eldad Klaiman**[1,*]

[1]Roche Diagnostics GmbH, [2]Roche Diagnostics S.L.
[3]Technical University of Munich, [4]Helmholtz Munich

## Abstract

Training multimodal foundation models is challenging due to the limited availability of multimodal datasets. While many public datasets pair images with text, few combine images with audio or text with audio. Even rarer are datasets that align all three modalities at once. Critical domains such as healthcare, infrastructure, or transportation are particularly affected by missing modalities. This makes it difficult to integrate all modalities into a large pre-trained neural network that can be used out-of-the-box or fine-tuned for different downstream tasks. We introduce LoReTTa (**L**inking m**O**dalities with a t**R**ansitive and commutativ**E** pre-**T**raining s**T**r**A**tegy) to address this understudied problem. Our self-supervised framework unifies causal modeling and masked modeling with the rules of commutativity and transitivity. This allows us to transition within and between modalities. As a result, our pre-trained models are better at exploring the true underlying joint probability distribution. Given a dataset containing only the disjoint combinations $(A, B)$ and $(B, C)$, LoReTTa can model the relation $A \leftrightarrow C$ with $A \leftrightarrow B \leftrightarrow C$. In particular, we show that a transformer pre-trained with LoReTTa can handle any mixture of modalities at inference time, including the never-seen pair $(A, C)$ and the triplet $(A, B, C)$. We extensively evaluate our approach on a synthetic, medical, and reinforcement learning dataset. Across different domains, our universal multimodal transformer consistently outperforms strong baselines such as GPT, BERT, and CLIP on tasks involving the missing modality tuple.

## 1 Introduction

Integrating multiple modalities, such as images, text, or speech, promises to provide a more comprehensive and holistic understanding of otherwise intractable problems. By leveraging the unique and complementary strengths of each data type, this approach results in a more synergistic representation of the underlying phenomena – enabling deep learning systems to excel and adapt to real-world scenarios. Thus, there has been impressive progress in the field of multimodal learning in recent years. While most research focuses on multimodal datasets where all data points have all modalities, there has also been work on missing modalities [44, 45]. However, all these models and methods rely heavily on at least a subset of samples where all modality combinations are still present (Figure 1). In practice, it is not easy to obtain such datasets when we consider more than two modalities, e.g. three modalities. In the worst case, we might end up with a dataset containing only modalities $A$ and $B$, and another disjoint dataset containing modalities $B$ and $C$. Thus, the combination $A$ and $C$

---

[*]Equal contribution.

37th Conference on Neural Information Processing Systems (NeurIPS 2023).

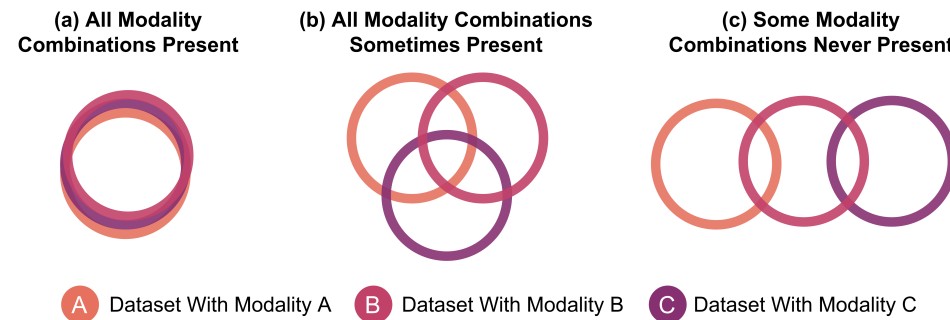

**(a) All Modality Combinations Present**

**(b) All Modality Combinations Sometimes Present**

**(c) Some Modality Combinations Never Present**

Ⓐ Dataset With Modality A  Ⓑ Dataset With Modality B  Ⓒ Dataset With Modality C

Figure 1: Venn diagrams showing the relationship between datasets with different modalities $A$, $B$, and $C$. Overlapping datasets indicate that the dataset contains samples with aligned modalities (i.e., audio, image, and text files belonging to the same concept). While recent work has mostly focused on datasets where at least some samples have all modality combinations available (a, b), we investigate the case where some modality combinations, e.g. $(A, C)$ and $(A, B, C)$, are missing entirely (c).

is unconditionally missing. So we asked ourselves, "Is it possible to pre-train a multimodal neural network that can handle any combination of modalities at inference time? Optimally, the model will achieve better downstream performance the more modality combinations are given. This includes the never-seen combinations $(A, C)$ and $(A, B, C)$.

To the best of our knowledge, this scenario has never been considered in the literature. We propose a novel transitive pre-trained transformer (Figure 2) that can deal with such situations. LoReTTa (**L**inking m**O**dalities with a t**R**ansitive and commutativ**E** pre-**T**raining s**TrA**tegy) can link modalities and transition between them using generative pre-training (predicting the next token), the commutative property $(A, B) = (B, A)$, and the transitive relation $(A \rightarrow B) \wedge (B \rightarrow C) \Rightarrow (A \rightarrow C)$. We show theoretically and experimentally that by learning the relationship within and between tokens from different data distributions, LoReTTa is well suited for training multimodal models.

This has far-reaching implications for safety-critical domains such as healthcare, infrastructure, and transportation. In medicine, for example, it is already difficult to obtain a large dataset with a single modality to train powerful foundation models – such as those used in general computer vision and natural language processing. Finding a dataset with two or more modalities is even more challenging. But combining them is important because it is common practice to look at different types of data, such as radiological images, genetic sequences, and microscopic slides, to determine the optimal treatment. Similarly, infrastructure monitoring models or automated driving systems require input from a variety of sensors. However, not all modality combinations might be present in the same training set. With LoReTTa, we demonstrate a unique way to address this challenging and under-researched problem.

## 2 Related work

With the introduction of transformers [63], the fields of natural language processing (NLP) and computer vision (CV) have experienced a significant shift in performance and capabilities. These models excel at a wide range of complex tasks, including machine translation, text summarization, question answering, sentiment analysis, natural language generation, image classification, image captioning, semantic segmentation, object detection, and face recognition – achieving state-of-the-art performance in each area [40]. Transformers have also opened up new possibilities for multimodal learning due to their lack of inductive bias. They can process and fuse different types of data, such as audio, image, video, and text, using self- and cross-attention mechanisms. As a result, transformers are considered to be universal, modality-agnostic learners [43, 22].

However, their lack of inductive bias comes at a price. Unlike other neural network architectures, transformers make few assumptions about the structure of the data [18]. This makes them highly expressive, but they also require a large amount of data to learn effectively and generalize well [28]. Consequently, self-supervised learning (SSL) on large amounts of unlabeled data has been proposed for transformers. In NLP, this strategy has been very successful, with methods such as masked

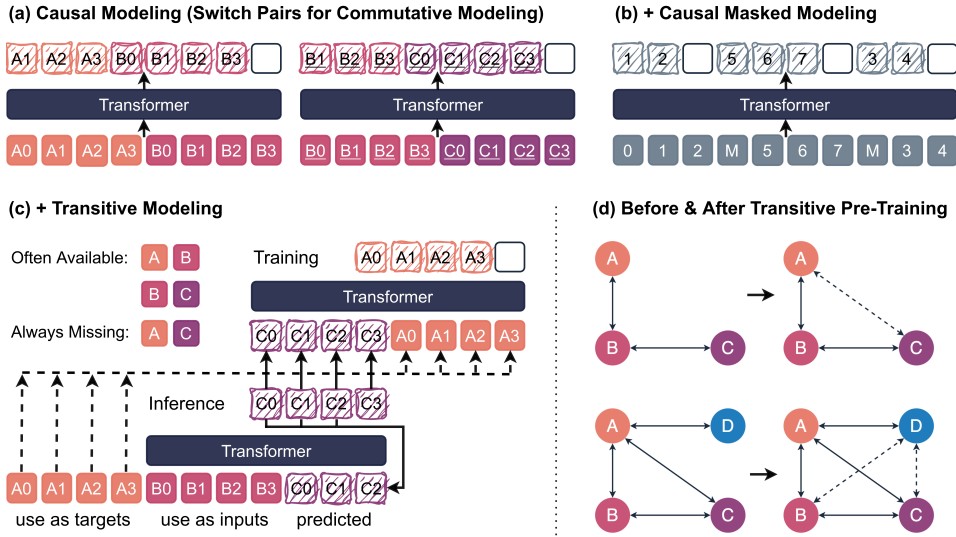

Figure 2: LoReTTa consists of two novel self-supervised strategies: commutative and transitive modeling. (a) In commutative modeling, we apply causal modeling in a commutative manner to generate modality $A$ from $B$ and modality $B$ from $A$ – given the aligned input sample $(A, B)$. For the aligned but disjoint data point $(B', C)$, we apply the same technique. (b) To ensure that the model learns bidirectional relations between the input tokens, we apply a modified variant of generative pre-training called causal masked modeling. (c, d) Next, we use transitive pre-training to learn any missing conditional joint distributions. The idea is simple. We randomly select a sample and use the linking modality $B$ to predict the missing modality $C$, which is then used to reconstruct the existing modality $A$. The last step is crucial because it ensures that all modalities are properly aligned.

modeling and causal modeling used to train large language models (LLMs) such as BERT [23] and GPT [49] variants [42, 46].

Recently, attempts have been made to unify both approaches through frameworks like prefix modeling [53], permutation modeling [71], causal masked modeling [1], or unified language learning [59]. When applied to image classification, token-based pre-training methods [8, 15] have demonstrated their ability to outperform established computer vision approaches such as SimCLR [17] and DINO [13]. As a result, these sequence modeling techniques have been successfully adapted and extended to a wide range of domains, encompassing programming [16], biochemistry [41], and reinforcement learning [35].

Adding or combining different modalities to the training process could open up three possibilities: positive transfer, cross-modal understanding, and cross-modal generation. Positive transfer occurs when learning in one modality helps to improve performance in another modality. For example, it has been shown that aligning images with text leads to better classification performance on ImageNet [51]. Cross-modal understanding refers to the explanation of one modality in terms of the other. Models with this capability can describe an image using text [4] or infer protein structures from sequences [30]. With cross-modal generation, one can even generate one modality using the other. Examples are text-to-image generation [54] and text-to-music generation [3]. Many modern multimodal foundation models are based on either contrastive learning [26], masked modeling [66], or causal modeling [72].

In this paper, we present a novel pre-training method that combines recent advances in self-supervised learning into a unified framework. LoReTTa uses causal modeling for generative tasks and masked modeling for discriminative tasks. It extends both approaches by using commutative and transitive modeling to seamlessly integrate and impute modalities. This allows us to learn expressive features and relationships within multimodal datasets. The pre-trained models can be used for various downstream tasks such as infilling, classification, survival prediction, or cross-modal generation. The most similar to our work is the RNN-based model MCTN [47]. However, it can only be used as a classifier, does not model all combinations of modalities, and requires all modalities to be aligned and present at training time. LoReTTa, on the other hand, does not need such strong assumptions.

# 3 Methods

**Model architecture:** LoReTTa (Figure 2) uses the autoregressive transformer decoder architecture described in Radford et al. [50]. This model includes a modified initialization, pre-normalization, and GELU activation. In addition, we choose RMSNorm [74] over LayerNorm [7], as in Rae et al. [52] and Brown et al. [12]. We also replace the StandardAttention [63] with FlashAttention [21] to reduce memory usage and runtime – similar to Touvron et al. [61]. Following Chowdhery et al. [19], we train without dropout and do not use biases in any of the dense kernels or layer norms. While any general sequence model can work for next token prediction, we choose the transformer [63] in particular for its simplicity and scalability. Theoretically, however, our method also works with other backbones, such as the recently proposed RWKV [10] and Hyena [48].

**Data representation:** Since transformers take a sequence of tokens as input, we need to tokenize our data first. We use common methods such as raw byte encoding [56, 29, 15], learned lookup tables [33, 58, 57], and vector quantization [62, 55, 54, 3]. After tokenization, we use a parameterized embedding function to project all tokens into a common feature space. In addition, we add modality-specific tokens to indicate the beginning and end of a modality. We also learn modality-specific absolute position encodings that are added to the feature vectors. The representations for each modality are then concatenated, but only if they belong to the same object/subject. For example, if we have an image-audio pair or an image-text pair describing the same concept, we would concatenate their token sequences and feed them into the transformer. However, if only one modality is available, we use only that modality. In summary, our pre-processing step is described below:

- We tokenize and project all samples $a, b, c$ into a common vector space, obtaining the token embeddings $[a_1, \ldots, a_l], [b_1, \ldots, b_m]$, and $[c_1, \ldots, c_n]$.
- If $a$ and $b$ are aligned, then we concatenate them as $s = [a_1, \ldots, a_l; b_1, \ldots, b_m]$ or $s = [b_1, \ldots, b_m; a_1, \ldots, a_l]$, which ensures commutativity.

**Pre-training strategy:** We train the transformer using causal language modeling. Given a sequence of tokens $s_{1:L}$ and parameters $\theta$, the data is modeled using the chain rule of probability

$$\log p_\theta(s_1, \ldots, s_L) = \sum_{l=1}^{L} \log p_\theta(s_l | s_1, \ldots, s_{l-1}) \tag{1}$$

Thus, the task of the model is to predict the next token given the sequence of previous tokens. As shown in GATO [56] and PALM-E [24], this objective is modality agnostic. The systems even learn rich internal world models [38, 36]. We ensure commutativity by randomly mixing the order of concatenation when appending tokens from two modalities to an input sequence, as described above. This allows the model to infer modality $A$ from $B$ and modality $B$ from $A$. We call this technique commutative modeling. Interestingly, this simple modification has not been used in recent multimodal foundation models.

Since generative pre-trained transformers only consider information from the left and ignore information from the right, they are known to lack bidirectional context [65], which is crucial for many downstream tasks [53]. Thus, we combine masked modeling (GPT-style models) and causal modeling (BERT-style models) into one framework. This recently developed approach is known in the literature under different names: fill-in-the-middle (FIM) [9], causal masked modeling (CM3) [1, 25], hybrid unidirectional modeling (HybUni) [5], span corruption + language modeling (SCLM) [59]. Conceptually, the idea is to randomly mask out a span (or spans) of tokens, replace them with sentinel tokens, and move the original tokens to the end. This change does not alter the training process based on Equation 1; however, it now allows the model to use the information to the right of the mask to predict the masked positions, which is not possible with traditional unidirectional causal modeling. The hybrid view not only leads to better performance in downstream tasks after fine-tuning but also retains the utility of language modeling [5, 59]. With CM3Leon [73], causal masked modeling has been scaled up to large vision-language models capable of generating and filling both text and images.

**Transitive modeling:** We now extend several language learning paradigms (masked, prefix, causal, causal masked) with our new technique, which we call transitive modeling. To do this, we return to the problem formulation outlined in Section 1. In particular, it requires us to combine modalities $A$ and

$C$, given only the disjoint combinations $(A, B)$ and $(B', C)$. We propose to randomly sample data points $(a, b), (b, a), (b', c)$, or $(c, b')$ during training and apply transitive modeling. For simplicity, we will consider the pair $(a, b)$. Since the model is trained to predict modality $C$ from $B$, we can infer the pseudo data point $\hat{c}$ from $b$. Now we train the transformer to predict point $\hat{a}$ using the newly inferred sample $\hat{c}$ in a way that minimizes the loss with respect to the original sample $a$, ensuring that all our modalities are aligned (Figure 2c). In this way, we model the conditional distribution of the missing pair $(C, A)$ – which, due to commutativity, is equal to $(A, C)$. This idea has some conceptual similarities to cycle consistency. However, our method is much more general since it can be applied to many combinations of modalities as long as there is a linking modality, see Figure 2d.

In fact, upon closer inspection, transitive modeling is quite different from cycle consistency. The most popular version of cycle consistency was introduced in CycleGAN [75]. Given an input $a$ in domain/modality $A$, one wants to generate an aligned output $b$ in domain/modality $B$. The model then computes a discriminative loss $D$ on $b$ and a reconstruction loss $L$ on the original $a$ and the predicted $\hat{a}$. A different version of cycle consistency was used in MCTN [47]: Given aligned inputs $(a, b)$ from modalities $A$ and $B$, $a$ is used to predict $\hat{b}$, and the predicted $\hat{b}$ is used to predict $\hat{a}$. Then the reconstruction loss $L$ is applied to compare $a$ and $\hat{a}$ as well as $b$ and $\hat{b}$. Transitive modeling, on the other hand, starts with the aligned modalities $(a, b)$ and $(b', c)$, it uses $b$ to predict $\hat{c}$ and $\hat{c}$ to predict $\hat{a}$, the reconstruction loss then compares $a$ and $\hat{a}$. Since we use commutative modeling, the other direction starting from a different sample $(b', c)$ is also learned. In summary, we have the following three approaches:

- CycleGAN: Given $a$, model $a \to \hat{b} \to \hat{a}$ and calculate $L(a, \hat{a}) + D(\hat{b})$.
- MCTN: Given $(a, b)$, model $a \to \hat{b} \to \hat{a}$, and calculate $L(a, \hat{a}) + L(b, \hat{b})$.
- LoReTTa: Given $(a, b)$, model $b \to \hat{c} \to \hat{a}$ and calculate $L(a, \hat{a})$.

Thus, transitive modeling does not simply add another step to cycle consistency by extending $a \to \hat{b} \to \hat{a}$ to $a \to \hat{b} \to \hat{c} \to \hat{a}$. In fact, we model $b \to \hat{c} \to \hat{a}$. Intuitively, this ensures that the model does not "cheat" by memorizing the input $a$ in order to reconstruct $a$. We also do not use a cycle loss, because $a$ is not used as an input and output. Thus, our loss $L(a, \hat{a})$ is not cyclic with respect to the model's inputs. However, we still make sure that the predicted modality is consistent with the data as a whole. This has a similar flavor to cycle consistency, but as can be seen above, it is not the same. In summary, LoReTTa uses masked modeling to transition within modalities, causal modeling to transition between modalities, and vice versa thanks to commutative modeling. Moreover, not all modality combinations need to be available at training time, as we use transitive modeling to predict the missing modalities and connect them to the original data. In our experiments, we show that this approach allows us to learn very rich multimodal feature representations.

## 4 Theoretical analysis

We consider a dataset with modalities $X_1, ..., X_N$ to be aligned if there exists a subset $S$ sampled from a joint probability distribution $P(X_1, ..., X_N; Y)$ [37]. Here, $Y$ represents the label space. This form of alignment, where only labels are shared across modalities, is called weak alignment [14]. Conversely, when there is a functional relationship at the level of physical objects, we consider the alignment to be strong [14]. The definition of strong and weak alignment also extends to datasets where we do not have all modality combinations at once. For example, during data collection, we may recover only the paired samples $(x_1, x_2)$ and $(x_2', x_3')$, where $x_2$ and $x_2'$ do not belong to the same object/subject. Thus, we have datasets of the form $(X_1, X_2)$ and $(X_2, X_3)$ instead of $(X_1, X_2, X_3)$. This is different from the case where the dataset still contains some samples of the form $(x_1, x_2, x_3)$, which was considered in previous work. To model the unseen relationship $(X_1, X_3)$ in a self-supervised way, we cannot simply use techniques like masked modeling or causal modeling. The neural network would have to implicitly learn

$$p(x_3|x_1) = \frac{p(x_1, x_3)}{p(x_1)}, \tag{2}$$

by deriving $p(x_1, x_3) = \sum_{x_2} p(x_1, x_2, x_3)$ and $p(x_1, x_2, x_3) = p(x_3|x_2, x_1)p(x_2|x_1)p(x_1)$ from the chain rule of probability. Empirically [6, 39, 69], both the unidirectional and bidirectional

Table 1: Perplexity on the SVL-MNIST test set. The subscript describes the pre-training datasets with image (I), text (T), or speech (A), while the blue colors highlight the unseen modality pairs.

| Training | Testing | | | | | |
|---|---|---|---|---|---|---|
| | **I** | **T** | **A** | **(I, T)** | **(I, A)** | **(T, A)** |
| **CM3$_{(I)}$** | 1.55 | - | - | - | - | - |
| **CM3$_{(T)}$** | - | 2.05 | - | - | - | - |
| **CM3$_{(A)}$** | - | - | 18.63 | - | - | - |
| **C2M3$_{(I,T)}$** | 1.62 | 2.28 | - | 1.76 | - | - |
| **C2M3$_{(I,A)}$** | 1.61 | - | 20.09 | - | 4.42 | - |
| **C2M3$_{(T,A)}$** | - | 2.25 | 19.85 | - | - | 10.68 |
| **C2M3$_{(I,T), (I,A)}$** | 1.64 | 2.19 | 21.37 | 1.76 | 4.39 | **30.88** |
| **C2M3$_{(T,I), (T,A)}$** | 1.65 | 2.18 | 19.69 | 1.76 | **104.27** | 11.26 |
| **C2M3$_{(A,I), (A,T)}$** | 1.64 | 2.19 | 23.45 | **9.70** | 4.66 | 11.92 |
| **LoReTTa$_{(I,T), (I,A)}$** | 1.56 | 2.17 | 18.77 | 1.72 | 3.14 | **11.21** |
| **LoReTTa$_{(T,I), (T,A)}$** | 1.59 | 2.10 | 18.95 | 1.70 | **5.04** | 10.03 |
| **LoReTTa$_{(A,I), (A,T)}$** | 1.60 | 2.14 | 20.35 | **3.13** | 4.36 | 10.65 |
| **C2M3$_{(I, T, A)}$** | 1.55 | 2.35 | 19.43 | 1.72 | 4.25 | 10.55 |

approaches fail this task even within one modality because it is difficult to model $p(x_3|x_2, x_1)$ due to the missing set $(X_1, X_2, X_3)$. This is where our proposed transitive modeling paradigm comes in.

Suppose we have a dataset of the form $(X_1, X_2), (X_2, X_3), \ldots, (X_{N-1}, X_N)$, where $X_k$ and $X_l$ are not aligned when $k + 1 \neq l$. By applying causal modeling to all pairs, we can learn the transition $X_1 \rightarrow X_2 \rightarrow \cdots \rightarrow X_{N-1} \rightarrow X_N$. Adding our proposed commutative modeling, we get an even stronger bidirectional relation: $X_1 \leftrightarrow X_2 \leftrightarrow \cdots \leftrightarrow X_{N-1} \leftrightarrow X_N$. Now, let us interpret each modality as a node on a high-dimensional graph. This gives us a connected graph (minimum spanning tree) where we can walk from one modality to any other modality. Thus, given a real data point $x_i$, we can use transitive modeling to sample pseudo data points $x_j$ by transitioning from $x_i$ to $x_j$. Having sampled many pairs $(x_i, x_j)$, we get a new dataset $(X_i, X_j)$. It can be used again in the generative pre-training process to learn the new transition $X_i \leftrightarrow X_j$. We apply this idea to all $i$ and $j$. In the end, we learn the conditional distribution between all samples and obtain a fully connected graph. This idea holds for more complicated modality combinations as long as there is a bridging modality, i.e., we have a spanning tree (see Figure 2d). We can also sample all missing tuples $(x_k, ..., x_l)$. Thus, LoReTTa allows us to approximate any missing joint probability distribution $P(X_k, ..., X_l)$ by sampling and training with the missing modality combinations.

Ideally, there would be no generalization error after pre-training. In practice, however, we would not be able to accurately recover the ground truth of the missing modality if it were available. For the datasets $(X_1, X_2)$ and $(X_2, X_3)$, the learned generative model $f$ would make the following errors:

$$f(x_1|x_2) = x_1 + e_{2,1}, \quad f(x_2|x_1) = x_2 + e_{1,2}, \quad f(x_2|x_3) = x_2 + e_{3,2}, \quad f(x_3|x_2) = x_3 + e_{2,3} \tag{3}$$

where all $e_{i,j}$ are vector-valued error terms. Without loss of generality, we do not keep track of the sign of the error, since we can always write $e'_{i,j} = -1 \cdot e_{i,j}$. Assuming that $f$ is differentiable and using the multivariate Taylor expansion, we obtain

$$x_3 = f(x_3|f(x_2|x_1) + e_{1,2}) + e_{2,3} \approx f(x_3|f(x_2|x_1)) + J_f(x_3|f(x_2|x_1))J_f(x_2|x_1)e_{1,2} + e_{2,3}, \tag{4}$$

where $J_f$ is the Jacobian of $f$. By iteratively applying the Taylor rule, a more general error propagation formula is obtained for longer chains. Intuitively, errors from the first transitions are amplified more and more. However, because we use the predicted $x_3$ to reconstruct $x_1$, we can actually bound the error term $e_{1,3} := J_f(x_3|f(x_2|x_1))J_f(x_2|x_1)e_{1,2} + e_{2,3}$. Recall that we have a ground truth $x_1$ for which the following is true

$$x_1 = f(x_1|x_3) + e_{3,1} = f(x_1|f(x_3|x_2) + e_{2,3}) + e_{3,1}$$
$$= f(x_1|f(x_3|f(x_2|x_1) + e_{1,2}) + e_{2,3}) + e_{3,1} \tag{5}$$

Table 2: After pre-training, a classifier is trained on the frozen models. We analyze the classification accuracy in a few-shot scenario with 100 samples per class over three trials on SVL-MNIST. The subscript denotes the pre-training datasets with image (I), text (T), or speech (A) modalities, while the blue colors highlight the results on the unseen modality tuple during pre-training.

| Training | Testing | | | | | | |
|---|---|---|---|---|---|---|---|
| | I | T | A | (I, T) | (I, A) | (T, A) | (I, T, A) |
| $CM3_{(I)}$ | $79.7_{\pm0.8}$ | - | - | - | - | - | - |
| $CM3_{(T)}$ | - | $57.0_{\pm0.5}$ | - | - | - | - | - |
| $CM3_{(A)}$ | - | - | $80.3_{\pm0.6}$ | - | - | - | - |
| $C2M3_{(I,T)}$ | $82.9_{\pm0.6}$ | $61.9_{\pm0.7}$ | - | $89.7_{\pm0.2}$ | - | - | - |
| $C2M3_{(I,A)}$ | $83.8_{\pm0.5}$ | - | $79.9_{\pm0.2}$ | - | $89.9_{\pm0.1}$ | - | - |
| $C2M3_{(T,A)}$ | - | $61.1_{\pm0.4}$ | $79.5_{\pm0.2}$ | - | - | $85.2_{\pm0.3}$ | - |
| $C2M3_{(I,T),\,(I,A)}$ | $83.1_{\pm0.8}$ | $59.8_{\pm0.4}$ | $80.7_{\pm0.4}$ | $87.9_{\pm0.6}$ | $89.3_{\pm0.2}$ | $71.2_{\pm0.4}$ | $87.3_{\pm0.4}$ |
| $C2M3_{(T,I),\,(T,A)}$ | $81.7_{\pm0.1}$ | $63.2_{\pm0.6}$ | $78.5_{\pm0.6}$ | $89.1_{\pm0.3}$ | $72.9_{\pm0.2}$ | $84.8_{\pm0.5}$ | $83.2_{\pm0.4}$ |
| $C2M3_{(A,I),\,(A,T)}$ | $80.8_{\pm0.4}$ | $63.9_{\pm0.6}$ | $82.8_{\pm0.7}$ | $77.8_{\pm0.5}$ | $89.8_{\pm0.1}$ | $88.1_{\pm0.5}$ | $89.9_{\pm0.5}$ |
| $LoReTTa_{(I,T),\,(I,A)}$ | $82.7_{\pm0.9}$ | $62.8_{\pm0.4}$ | $82.5_{\pm0.6}$ | $88.5_{\pm0.7}$ | $89.7_{\pm0.1}$ | $84.0_{\pm0.2}$ | $90.7_{\pm0.6}$ |
| $LoReTTa_{(T,I),\,(T,A)}$ | $81.2_{\pm0.9}$ | $63.3_{\pm0.4}$ | $81.0_{\pm0.5}$ | $89.0_{\pm0.3}$ | $80.9_{\pm0.6}$ | $85.5_{\pm0.5}$ | $87.8_{\pm0.3}$ |
| $LoReTTa_{(A,I),\,(A,T)}$ | $80.1_{\pm0.3}$ | $62.1_{\pm0.6}$ | $84.2_{\pm0.5}$ | $83.0_{\pm0.7}$ | $90.4_{\pm0.3}$ | $89.0_{\pm0.2}$ | $91.6_{\pm0.7}$ |
| $C2M3_{(I,T,A)}$ | $80.3_{\pm0.4}$ | $58.3_{\pm0.4}$ | $79.3_{\pm0.7}$ | $85.2_{\pm0.2}$ | $85.3_{\pm0.4}$ | $82.5_{\pm0.3}$ | $88.5_{\pm0.2}$ |

Since we are minimizing the reconstruction error of $x_1$ given $\hat{x}_3 = x_3 + e_{1,3}$ during backpropagation, the error terms $e_{1,2}$ and $e_{2,3}$ must be bounded. And since they make up the error term $e_{1,3}$, it too must be bounded, as long as $f$ and $J_f$ are.

# 5 Experimental evaluation

**Evaluation protocol:** We empirically analyze LoReTTa on a constructed dataset, a real-world medical dataset, and an offline reinforcement learning dataset with three modalities. For the downstream tasks, we choose object classification, survival prediction, and cross-modal translation, respectively. With these three tasks, we cover both discriminative and generative problems. We assess the former through linear probing and the latter in a zero-shot scenario. This allows us to directly evaluate the quality of the learned features. The order of the experiments is as follows: First, we perform an ablation study on the synthetic dataset to analyze the effect of transitive modeling on different modality combinations. Second, we compare LoReTTa with other methods such as masked (BERT), causal (GPT), and contrastive (CLIP) modeling on the real-world medical dataset. Third, we compare our model with a state-of-the-art autoregressive cross-modal transformer on the offline gaming dataset. In the Appendix, we list all the optimization hyperparameters and model configurations. We also publish the pseudocode and data processing pipeline.

**SVL-MNIST:** We test our method on a custom dataset derived from real-world data that includes speech (A), vision (I), and language (T) modalities representing 10 different categories. The speech dataset features about 40,000 spectrograms from AudioMNIST [31], the vision dataset comprises 70,000 images from MNIST [34], and the language dataset consists of 130,000 documents from WineReviews [60]. We link the datasets via their labels 0-9. This approach results in weakly aligned modalities. For the sake of clarity, we will refer to this dataset as Speech-Vision-Language-MNIST (SVL-MNIST). We randomly split the dataset such that they have exactly the same relationship, as shown in Figure 1c. In particular, our bimodal datasets (A, I), (A, T), and (I, T) each have 12,000 non-overlapping samples. All remaining samples are part of the unimodal datasets A, I, and T.

**TCGA-OMICS:** The medical dataset contains omics (e.g., genomics, proteomics, and transcriptomics) sequencing values from The Cancer Genome Atlas Program (TCGA) [67]. We select the three largest subsets, which include messenger ribonucleic acid (mRNA), micro-ribonucleic acid (miRNA), and reverse-phase protein array (RPPA) from up to ∼11,000 patients. We align the dataset at the patient level and obtain approximately 7,000 patients with all three modalities. To simulate a setting with missing modality combinations, we split the dataset into two subsets (mRNA, miRNA) and (mRNA, RPPA) with non-overlapping 3,500 samples each. We choose mRNA as the linking modality because it is one of the most widely available genetic modalities. TCGA is quite unique in that it is the only publicly available medical dataset with multiple aligned modalities.

**MUGEN-GAME:** As a large-scale and multimodal dataset, we choose MUGEN-GAME [27] with 375,000 naturally aligned video (V), audio (A), and text (T) samples collected by a reinforcement learning agent in the closed-world platform game CoinRun. Mugen, the main character, walks, jumps, collects coins, kills monsters, climbs ladders, and dies – triggering various sound effects that are overlaid with background music. All of their adventures are recorded in three-second video clips and audio tracks. A group of human narrators provide rich text descriptions for each scene. We use the most important modality in video games, video, as the linking modality and consider the disjoint datasets (V, A) and (V, T) in our final experiments.

**Implementation details:** For the first two experiments, we use a transformer decoder with 8 layers and 8 attention heads, with an embedding dimension of 512 and query, key, and value dimensions of 64 each. This compact architecture has proven effective in various tasks and has been widely used in the literature [23, 28, 36]. We apply a single parameterized embedder to encode tokens and add modality-specific learned position embeddings [54, 56, 2]. For optimization, we choose the AdamW algorithm with a learning rate of 6e-4, a weight decay factor of 0.1, and a gradient clipping of 1. The learning rate undergoes a 10-fold decay using cosine annealing and a linear warm-up during the first couple hundred steps. For the third experiment, we use exactly the same training hyperparameters and model configurations as in the baseline [27]. In particular, the transformer has 12 layers, 8 attention heads, and an embedding dimension of 768. The query, key, and value dimensions are each 96.

**Tokenization scheme:** Due to the low dimensionality of the SVL-MNIST dataset, we directly encode the raw byte streams by binning the continuous values to a dictionary of 256 each. TCGA-OMICS data can be very high-dimensional and must be handled differently. For example, an mRNA expression array typically contains more than 20,500 entries. Therefore, we first reduce the dimensions by a factor of 10 using PCA and then bin the values to a dictionary of 1,000 (similar to recent work [11, 70, 20]). The final input sizes are 2,048, 64, and 16, respectively – retaining their original relative lengths. The high-dimensional video and audio samples in MUGEN-GAME are tokenized identically to Hayes et al. [27] using their pre-trained VQ-VAE encoders.

## 6 Results

### 6.1 SVL-MNIST

In the first experiment, we train one model for each modality (A, I, and T) using causal masked modeling (CM3). We then train four additional models using commutative causal masked modeling (C2M3) on the bimodal datasets (A, I), (A, T), and (I, T), as well as the trimodal dataset (A, I, T). These seven models serve as unimodal, multimodal, and "upper" baselines, respectively. Then, using C2M3 with T as the linking modality, we train simultaneously on both (I, T) and (A, T), and repeat this process for all modality combinations. Next, we initialize LoReTTa with the C2M3 model weights to explicitly merge the bimodal datasets via the linking modality. After pre-training, we freeze all models and evaluate them in a low-data regime with only 1,000 labeled samples (100 per class). We perform three runs, each with a newly randomized subset, and report the classification accuracy along with the perplexity on the test set. Note that we only train the models on the modalities that the model saw during pre-training. This also means that during probing, the linear classifier will see some samples of the unseen modality pair. However, the pre-trained backbone itself did not see them. In this way, we can analyze how the model combines previously unseen mixture of input modalities (modality combinations).

When pre-trained only on the pairs (T, I) and (T, A), LoReTTa achieves a remarkably low perplexity of 5.04 on the unseen pair (I, A), as shown in Table 1, significantly outperforming the baseline non-transitive model trained with C2M3, which has a perplexity of 104.27. This trend extends to other combinations as well. For example, the model trained on (A, I) and (A, T) and evaluated on (I, T) has a perplexity of 3.13 (LoReTTa) compared to 9.70 (C2M3). Similarly, training on (I, T) and (I, A) and evaluating on (T, A) yields a perplexity of 11.21 (LoReTTa) versus 30.88 (C2M3). Importantly, LoReTTa consistently achieves similar perplexities for unseen and seen modality combinations, demonstrating its ability to effectively learn the missing distribution. A similar trend appears in the downstream task (Table 2). In almost all cases, LoReTTa consistently outperforms C2M3 on the unimodal datasets A, I, and T, as well as the bimodal datasets (A, I), (A, T), or (I, T). In a nutshell, if any of the modality pairs are missing, training a model with LoReTTa is recommended because it uniquely integrates these modalities and improves accuracy compared to any single-modality model.

Table 3: After pre-training, a Cox proportional hazards model is trained on the extracted features from TCGA-OMICS. We use mixtures of mRNA (M), miRNA (I), and RPPA (R) as inputs. Blue highlights the c-index on the unseen tuples, and the underline highlights the second-highest value.

| Training | Testing | | | | | | |
|---|---|---|---|---|---|---|---|
| | **M** | **I** | **R** | **(M, I)** | **(M, R)** | **(I, R)** | **(M, I, R)** |
| **BERT** | 0.592 | 0.621 | 0.594 | 0.595 | 0.613 | 0.594 | 0.610 |
| **T-BERT** | 0.573 | 0.618 | **0.626** | 0.580 | 0.591 | **0.623** | 0.608 |
| **GPT** | 0.575 | 0.590 | 0.611 | 0.585 | 0.576 | 0.606 | 0.588 |
| **T-GPT** | 0.616 | 0.607 | 0.599 | 0.620 | 0.619 | 0.611 | 0.622 |
| **CLIP** | 0.561 | 0.603 | 0.587 | 0.610 | 0.600 | **0.623** | 0.612 |
| **L-CLIP** | 0.588 | 0.615 | 0.587 | 0.620 | 0.599 | 0.614 | 0.620 |
| **C2M3** | 0.621 | 0.599 | 0.599 | 0.624 | 0.620 | 0.571 | 0.624 |
| **LoReTTa** | **0.652** | **0.623** | 0.563 | **0.660** | **0.652** | **0.623** | **0.657** |
| **C2M3**(M, I, R) | 0.665 | 0.635 | 0.645 | 0.643 | 0.671 | 0.650 | 0.620 |

In fact, providing a few samples with three modalities during linear probing further increases the classification score of models pre-trained with LoReTTa. Interestingly, the "upper" baseline trained on all aligned modalities often lags behind LoReTTa despite favorable perplexity scores. This strongly suggests some kind of negative transfer or modality competition that LoReTTa is able to overcome.

## 6.2 TCGA-OMICS

We pre-train our transformer with C2M3 and initialize LoReTTa with the weights of the C2M3 model, as we did in our SVL-MNIST experiments. For GPT, we simply use the algorithm of C2M3, but disable causal masking and commutative switching. BERT uses the same transformer architecture; we simply change the unidirectional attention mask to a bidirectional attention mask. To directly demonstrate the benefit of transitive modeling, we extend both GPT and BERT using this technique. The resulting models are named T-GPT and T-BERT. The CLIP-based model consists of 3 encoders (each using our same transformer architecture) and a contrastive loss [32] to align the features. After pre-training, we extract the embeddings and fit a Cox proportional hazards model with an elastic net penalty for each modality combination using all available labels. In 6 out of 7 test sets, the LoReTTa pre-trained transformer achieves the highest c-index (Table 3). Most importantly, this includes the unseen modality pair (miRNA, RPPA) and the triplet (mRNA, miRNA, RPPA). The only frameworks on par with LoReTTa are T-BERT and CLIP on the unseen dataset (miRNA, RPPA). On the unseen trimodal dataset, T-GPT and CLIP give good results but are far behind LoReTTa. Overall, transitive modeling mostly improves the performance of all models. Notably, LoReTTa and CLIP are the only strategies that consistently allow for positive or neutral transfer, which is not always the case with other frameworks. A major drawback of contrastive learning is that it requires a separate encoder for each additional modality. In addition, CLIP-based approaches only excel at large batch sizes. Here, we compare models with the same computational constraints. We spent our limited resources on additional computing capacity. This allows us to scale up the CLIP experiment by increasing the batch size by a factor of 5 and the number of training steps by a factor of 3. L-CLIP consistently improves the c-index of CLIP but requires 3x more VRAM than LoReTTa. In contrast to the SVL-MNIST experiments, the 3-modal "upper-bound" model suffers less from negative transfer or modality competition and achieves the best result for the majority of modality combinations.

## 6.3 MUGEN-GAME

Recently, there has been a growing interest in cross-modal generation. While much research has focused on text-to-video or video-to-text translation, other cross-modal tasks such as video-to-audio, audio-to-video, text-to-audio, or audio-to-text have not been investigated as thoroughly. In our large-scale experiment, we address the latter task. We use video as the linking modality and consider the disjoint datasets (video, audio) and (video, text). Splitting the training set equally gives us about 187,500 pairs in each set. We use the same optimization hyperparameters and model architecture as the state-of-the-art upper baseline MMGPT [27] to train our version of (MM)GPT and also LoReTTa. As seen in our TCGA-OMICS experiment, BERT, unsurprisingly [5, 64], fails to generate long-range coherent samples (see results with mRNA). Therefore, we do not use BERT in this experiment. We

Table 4: Both GPT and LoReTTa are trained on disjoint (video, audio) and (video, text) pairs on MUGEN-GAME. We then evaluate the models on the unseen task of audio captioning using BLEU4, METEOR, and ROUGE. MMGPTs represent the upper-bound models.

| Method | Train | Test | BLEU4 | METEOR | ROUGE |
|---|---|---|---|---|---|
| GPT | $A \rightarrow V, V \rightarrow T$ | $A \rightarrow T$ | 1.7 | 18.5 | 30.7 |
| LoReTTa | $A \leftrightarrow V \leftrightarrow T$ | $A \rightarrow T$ | **2.8** | **20.8** | **34.7** |
| MMGPT | $A \rightarrow T$ | $A \rightarrow T$ | 6.7 | 19.4 | 27.1 |
| MMGPT | $V \rightarrow T$ | $V \rightarrow T$ | 7.8 | 21.3 | 29.1 |

also avoid training a CLIP model, as this would require us to train a text diffusion model (or equivalent generator) on top of the pre-trained encoder, which is neither end-to-end nor straightforward. For GPT, we train the model to generate video from audio and text from video. In this way, we force the model to implicitly learn how to caption audio tracks. In the case of LoReTTa, we model the transition between audio and text directly through transitive modeling. Remarkably, GPT trained on two bimodal datasets outperforms the audio-to-text baseline in terms of ROUGE, showing strong positive transfer (Table 4). LoReTTa further improves on GPT and even rivals the video-to-text baseline w.r.t. METEOR and ROUGE. The BLEU4 values are low for both models. Since BLEU involves exact n-gram matching, these results are to be expected given the strict data constraints.

## 7    Discussion & Conclusion

With LoReTTa, we introduced an innovative self-supervised learning framework for multimodal integration that learns any missing joint conditional distribution given a linking modality. We demonstrated this theoretically and practically. In particular, our method combines the rules of commutativity and transitivity to model the multimodal datasets $(A, C)$ and $(A, B, C)$ given the two non-overlapping datasets $(A, B)$ and $(B, C)$. In fact, we showed that the extracted features of a transformer pre-trained with LoReTTa are very expressive for all modality combinations, including the unseen ones. While we only evaluated our algorithm on datasets with three modalities, it is much more general because it can be used with any number of modalities and any mixture of modalities. As long as there is a chain of aligned modalities in the dataset, LoReTTa is able to learn the transition $(X_i \rightarrow ... \rightarrow X_j), ..., (X_i \rightarrow ... \rightarrow X_k)$ to model the missing distribution $P(X_j, ..., X_k)$ (as shown in Figure 2d). To our knowledge, this scenario has never been considered in the literature without training an ensemble of contrastive models based on the ideas of CLIP – most notably Wav2CLIP [68] and its extension ImageBind [26].

**Broader impacts.** We believe that LoReTTa will serve as a basis for training very powerful multimodal generative and discriminative models that will be useful in many areas of research. While our method itself cannot be misused, future models trained with LoReTTa could be exploited to generate harmful content of higher quality. However, LoReTTa was designed to improve the quality of neural networks that contribute to society, for example by training more accurate multimodal systems that can help doctors and other practitioners in their daily work.

**Environmental impacts.** We trained all of our models on a single NVIDIA A100-SXM4-40GB GPU using PyTorch 2.0. Techniques such as mixed-precision training, FlashAttention, and TF32 helped us reduce memory usage and runtime. Pre-training a model with causal masked modeling took us about 12 hours on average. On the other hand, fine-tuning with LoReTTa took longer – about 3 days. This is because we had to autoregressively decode the predicted samples one token at a time.

**Limitations.** LoReTTa is best used in a setting where we are missing some modality combinations. This only works if there is at least one connecting modality, otherwise, our method cannot be applied. But what about datasets where some samples have all modality combinations? Can our method still be used? Yes, this is possible. Although we have not analyzed this scenario, LoReTTa is likely to provide benefits by augmenting those data points that are still missing the remaining modalities.

**Acknowledgements.** M.T. is supported by the Helmholtz Association under the joint research school "Munich School for Data Science - MUDS". There are no other funding or competing interests.

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
