# Appendix: Training Transitive and Commutative Multimodal Transformers with LoReTTa

**Manuel Tran**[1,3,4] **Yashin Dicente Cid**[2] **Amal Lahiani**[1] **Fabian J. Theis**[3,4]

**Tingying Peng**[4,*] **Eldad Klaiman**[1,*]

[1]Roche Diagnostics GmbH**,** [2]Roche Diagnostics S.L.
[3]Technical University of Munich, [4]Helmholtz Munich

## A   Pre-training

Models pre-trained with a language learning paradigm share a similar set of hyperparameters. This is consistent with current best practices for state-of-the-art (multimodal) foundation models based on transformers. More on this topic can be found in the literature [2, 3]. Specifically, we use the AdamW optimizer with betas of (0.9, 0.95), epsilon of 1e-8, weight decay of 0.1, and gradient clipping at 1.0. The learning rate starts at 1e-7, increases linearly to 6e-4, and gradually decays to 6e-5 according to a cosine schedule. To ensure that the model sees batches containing different modalities and modality combinations during training, we accumulate batches across different dataloaders. For CLIP, we use betas of (0.9, 0.98), epsilon of 1e-6, and weight decay of 0.2. The maximum learning rate starts at 5e-4 and ends at 5e-5. The list of batch sizes, warm-up steps, and total training steps for each experiment can be found in Table A1 and Table A2. The above setting applies to our SVL-MNIST and TCGA-OMICS experiments. The MUGEN-GAME experiments follow the exact hyperparameters as the reference [1] – except that we train for 142,000 steps.

Table A1: The batch size, number of warm-up steps, and number of total training steps for each model in the SVL-MNIST experiment.

| Method | Batch Size | Warm-up Steps | Total Steps |
|---|---|---|---|
| **CM2$_{(I)}$** | 64 | 1,000 | 23,500 |
| **CM2$_{(T)}$** | 64 | 1,000 | 16,300 |
| **CM2$_{(A)}$** | 64 | 1,000 | 13,400 |
| **C2M3$_{(I,T)}$** | 64 | 200 | 18,800 |
| **C2M3$_{(I,A)}$** | 64 | 200 | 18,800 |
| **C2M3$_{(T,A)}$** | 64 | 200 | 18,800 |
| **C2M3$_{(I,T), (I,A)}$** | 16 | 400 | 37,500 |
| **C2M3$_{(T,I), (T,A)}$** | 16 | 400 | 37,500 |
| **C2M3$_{(A,I), (A,T)}$** | 16 | 400 | 37,500 |
| **LoReTTa$_{(I,T), (I,A)}$** | 16 | 400 | 9,400 |
| **LoReTTa$_{(T,I), (T,A)}$** | 16 | 400 | 9,400 |
| **LoReTTa$_{(A,I), (A,T)}$** | 16 | 400 | 9,400 |
| **C2M3$_{(I, T, A)}$** | 16 | 400 | 92,000 |

---

[*]Equal contribution.

37th Conference on Neural Information Processing Systems (NeurIPS 2023).

Table A2: The batch size, number of warm-up steps, and number of total training steps for each model in the TCGA-OMICS experiment.

| Method | Batch Size | Warm-up Steps | Total Steps |
|---|---|---|---|
| **BERT** | 16 | 400 | 5,500 |
| **T-BERT** | 16 | 400 | 1,100 |
| **GPT** | 16 | 400 | 5,500 |
| **T-GPT** | 16 | 400 | 1,100 |
| **CLIP** | 16 | 400 | 1,200 |
| **L-CLIP** | 80 | 400 | 3,800 |
| **C2M3** | 16 | 400 | 5,500 |
| **LoReTTa** | 16 | 400 | 1,300 |
| **C2M3**$_{(M, I, R)}$ | 16 | 400 | 9,900 |

# B   Linear probing

In our SVL-MNIST experiments, we freeze the backbone and train a linear classifier on top. SGD is used as the optimizer with the Nesterov momentum set to 0.9. The initial learning rate is 0.1, but it gradually decreases to zero during training by cosine annealing. We do not use weight decay. The batch size is 16, and the number of epochs (based on the validation sets) is listed in Table A3. For the TCGA experiments, we fit a Cox proportional hazards model with an elastic net penalty to the extracted features. The weights for the regularization terms $L_1$ and $L_2$ are both set to 0.5. Training is stopped when one of the following criteria is met: tol=1e-7 or iter=100000. The underlying optimization algorithm is based on coordinate descent, which successively minimizes the objective function along one direction at a time. It is particularly effective for problems with many features.

Table A3: Number of training epochs to fit the linear classifier on the SVL-MNIST dataset for each dataset containing different modalities and modality combinations.

| Training | Testing | | | | | | |
|---|---|---|---|---|---|---|---|
| | I | T | A | (I, T) | (I, A) | (T, A) | (I, T, A) |
| **CM2**$_{(I)}$ | 100 | - | - | - | - | - | - |
| **CM2**$_{(T)}$ | - | 100 | - | - | - | - | - |
| **CM2**$_{(A)}$ | - | - | 100 | - | - | - | - |
| **C2M3**$_{(I,T)}$ | 100 | 100 | - | 100 | - | - | - |
| **C2M3**$_{(I,A)}$ | 100 | - | 100 | - | 100 | - | - |
| **C2M3**$_{(T,A)}$ | - | 100 | 100 | - | - | 100 | - |
| **C2M3**$_{(I,T), (I,A)}$ | 500 | 100 | 100 | 500 | 500 | 500 | 500 |
| **C2M3**$_{(T,I), (T,A)}$ | 500 | 100 | 100 | 500 | 500 | 500 | 500 |
| **C2M3**$_{(A,I), (A,T)}$ | 500 | 100 | 100 | 500 | 500 | 500 | 500 |
| **LoReTTa**$_{(I,T), (I,A)}$ | 500 | 100 | 100 | 500 | 500 | 500 | 500 |
| **LoReTTa**$_{(T,I), (T,A)}$ | 500 | 100 | 100 | 500 | 500 | 500 | 500 |
| **LoReTTa**$_{(A,I), (A,T)}$ | 500 | 100 | 100 | 500 | 500 | 500 | 500 |
| **C2M3 (3-modal)** | 500 | 100 | 500 | 500 | 500 | 500 | 500 |

# C   Datasets

We divide the SVL-MNIST dataset into training, validation, and test sets (Figure A1). The validation set is merged with the training set after hyperparameter search. To obtain the multimodal dataset with completely missing modality combinations, we consider the three datasets (I, T), (T, A), and (A, I). The first dataset (I, T) consists of 12,000 paired samples from MNIST and WineReviews. The second dataset (T, A) is similarly constructed and contains 12,000 paired samples from WineReviews and AudioMNIST. We take another 12,000 from AudioMNIST and combine them with 12,000 samples from MNIST to get (A, I). All remaining samples are part of the unimodal datasets I, T, and A. There is no dataset with three modalities (I, T, A) – except for testing. Note that none of the samples in the datasets overlap. That is, all datasets have the same relationship as in Figure 1c of the main paper.

The TCGA-OMICS dataset contains 11,069 mRNA, 10,824 miRNA, and 7,790 RPPA samples. We align the dataset at the patient level and obtain 7,030 data points with three modalities (Figure A2). 1,030 of these are used for testing. The remaining samples are part of the training and validation set (again, the validation set is merged with the training set after hyperparameter tuning). In particular, the training set consists of the (mRNA, miRNA) and (mRNA, RPPA) datasets, each consisting of 3,000 paired samples. We choose mRNA as the linking modality because it is usually the most abundant and common modality available in medical datasets.

MUGEN-GAME consists of 375,368 fully aligned samples. It is divided into a training, validation, and test set of sizes 349, 666, 12,851, and 12,851, respectively. We randomly pair video and audio as well as video and text files. This results in two disjoint datasets of size 174,833 each for training.

For a fair comparison, we use only the fully aligned subset of the test set to report the final results, since the other sets contain different samples and are of different sizes. However, we keep the data split with the individual unaligned modalities (right side of Figure A1 and Figure A2) to make the dataset more flexible for future experiments.

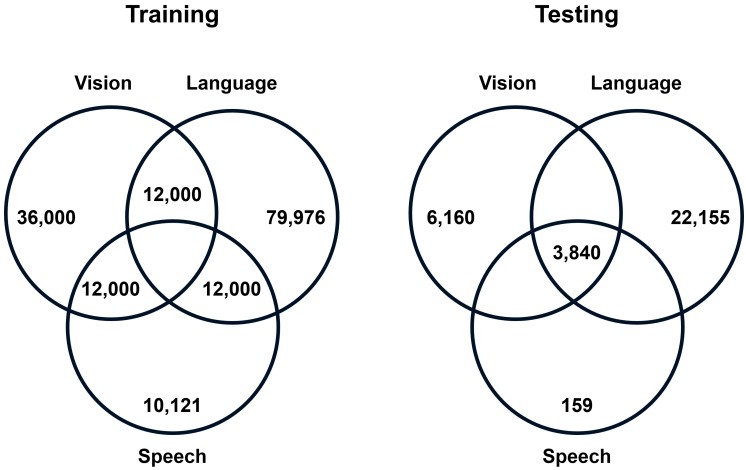

Figure A1: SVL-MNIST data split used for training and testing.

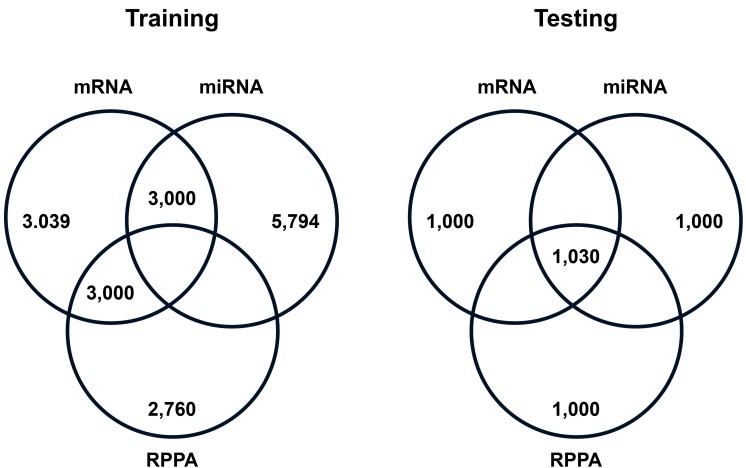

Figure A2: TCGA-OMICS data split used for training and testing.

# D  Pseudocode

The pseudocode below gives more insight into how LoReTTa pre-training is implemented. It outlines the main ideas and algorithmic steps to train a model using commutative and transitive modeling. Most importantly, the code shows how both are integrated into the causal modeling framework.

```python
class LoReTTa:
    """
    Pseudo-code for commutative and transitive modeling
    """
    def forward(self, tokens, modes=['commutative','transitive']):
        """
        tokens ... tokenized inputs, e.g., [x_0,...x_n, y_0,...,y_m]
        x_0, y_0 .... modality-specific tokens, 'a', 'b', or 'c'
        """

        if 'commutative' in modes: #shuffle modalities
            tokens = self.shuffle_modalities(tokens)

        if 'transitive' in modes: #generate missing modality
            existing_modalities = self.extract_modality_tokens(tokens)

            if ['a', 'b'] in existing_modalities: #case 1
                modality_a, modality_b = self.split_tokens(tokens)
                modality_c = self.model.generate([modality_b, 'c'])
                tokens = [modality_c, modality_a]

            if ['b', 'c'] in existing_modalities: #case 2
                modality_b, modality_c = self.split_tokens(tokens)
                modality_a = self.model.generate([modality_b, 'a'])
                tokens = [modality_a, modality_c]

            if ['a'] in existing_modalities \ #case 3
                and len(existing_modalities) == 1: #edge case with one modality
                modality_b = self.model.generate([modality_a, 'b'])
                modality_c = self.model.generate([modality_b, 'c'])
                tokens = [modality_c, modality_a]

            if ['b'] in existing_modalities \ #case 4
                and len(existing_modalities) == 1: #edge case with one modality
                modality_a = self.model.generate([modality_b, 'a'])
                modality_c = self.model.generate([modality_b, 'c'])
                tokens = self.shuffle_modalities([modality_a, modality_c])

            if ['c'] in existing_modalities \ #case 5
                and len(existing_modalities) == 1: #edge case with one modality
                modality_b = self.model.generate([modality_c, 'b'])
                modality_a = self.model.generate([modality_b, 'a'])
                tokens = [modality_a, modality_c]

            if self.prob_use_all_modalities < rand(1): #occcasionally use all modalities
                tokens = self.shuffle_modalities([modality_a, modality_b, modality_c])

        logits = self.model(tokens[:, :-1]) #get predictions
        targets = tokens[:, +1:] #shift targets

        loss = self.criterion(logits, targets) #calculate cce-loss
        return self.split_loss(loss) #return individual loss for each modality
```