# OpenReview forum: "Training Transitive and Commutative Multimodal Transformers with LoReTTa"
_NeurIPS.cc/2023/Conference — NeurIPS 2023 poster_

### Official Review · Reviewer_SR2A · 2023-06-28

**Soundness:** 4 excellent
**Presentation:** 4 excellent
**Contribution:** 4 excellent
**Rating:** 6
**Confidence:** 4

**Summary:**

This paper presents a learning paradigm to account for missing paired modalities during training. For example, for three modalities A, B, and C, the proposed system can be trained on (A, B) and (B, C) paired data but transfer to (A, C) or (A, B, C) paired data. To tokenize the modalities, the authors uses standard tokenizers for language, and raw bytes or spatially-reduced methods (CNN/VQ-VAE) to tokenize images and audios. The model is based on a standard transformer decoder, and the key novelties are: (a) commutative modeling by asking the model to next-token predict modality A from B, or B from A. (b) transitive modeling by producing pseudo data using a linking modality. To incorporate bidirectional context while using causal attention, casual masked modeling loss is employed. The proposed system is evaluated on both customized and real-world datasets and its performance is compared against popular multimodal learning paradigms including masked, casual, and contrastive objectives.

**Strengths:**

This paper conducts extensive experiments (section 5 and 6) on both constructed and real-world medical datasets with three modalities, and show that the proposed system is very effective when there is missing paired modality data. Especially, the authors perform careful ablations against mainstream popular pre-training objectives, including contrastive, casual, and masked modeling losses.

Theoretical analysis (section 4) based on perplexity is interesting and sound.

**Weaknesses:**

The proposed system should ideally be tested on larger-scale multimodal datasets, such as AudioSet [1] with image-text-audio modalities. Importantly, this would be crucial to see if the proposed system generalize to real-world vision-language-audio domains where the three modalities are naturally aligned. This setup would be different from the constructed SVL-MNIST dataset where the language modality (from WineReview) has weak or no correlation to the vision (MNIST) and audio (AudioMNIST) modalities.

For complete ablations, the authors could provide results on the “upper bound” training setups where:
- All samples have (A, B, C) modalities.
- There are (A, B), (B, C), (A, C) paired modality data.

Not a weakness but a suggestion:
L38: “To the best of our knowledge, this scenario has never been considered before in the literature.” —> There is a recent related work [2] in vision community that use images as the linking modality to bind all other modalities such as depth/audio/text.

[1] Audio set: An ontology and human-labeled dataset for audio events. Gemmeke et al. 2017.
[2] ImageBind: One Embedding Space To Bind Them All. Girdhar et al. 2023

**Questions:**

Some technical concerns that I have:
- The baseline linear probing accuracy on SVL-MNIST is rather poor. For instance, the vision-MNIST should at least have an accuracy above 95%, but the vision-only transformer in this paper only achieves ~80%.
- Also, the text modality of SVL-MNIST seems to be noisy as its linear probing accuracy is only 60%. Is the poor accuracy resulting from low-shot training with 100-shot per class, or due to some label noise?

**Limitations:**

Yes, the authors discussed limitations of their approach from both technical and societal perspectives.

---

> ### Author Rebuttal · Authors · 2023-08-07
>
> **Dear Reviewer SR2A,**
>
> We appreciate your time and effort in reading our manuscript and providing us with valuable feedback. It is wonderful to hear that the soundness, presentation, and contributions of our paper are considered excellent. We would like to address the remaining questions below.
>
> **Large-scale experiments.** We were faced with a limited computational budget. However, we were still able to perform extensive experiments on a synthetic and real medical dataset with meaningful applications. It is impressive to see that LoReTTa can effectively integrate highly complex omics data across different domains to improve survival predictions. The question, of course, is how to scale the model. According to the AudioSet website and paper, it is primarily a dataset for audio classification. The corresponding video clips must first be found and transcribed to obtain the video and text modalities. However, a recent video-language-audio multimodal dataset [Hayes et al., 2022] was recommended during the rebuttal. It contains 375,000 aligned data pairs sampled from a real-world reinforcement learning problem. Even with limited time and resources, we were able to finish the experiment and achieve performances that are competitive with the upper bound model (see global rebuttal). Because of this additional experiment, we could not provide the upper bound results for the other experiments. However, it was a reasonable decision to omit them. We considered a scenario where there is no dataset with all modalities present and aligned. Thus, in practice, we would not know the upper bound. The only relevant upper bound is the current state-of-the-art, which we have shown to outperform.
>
> **Linear probing accuracy.** Training on raw pixel values instead of image patches is more challenging since we lose potentially important spatial information [Chen et al., 2020; Jaegle et al., 2021; Yu et al., 2022]. We committed ourselves to this setting to make the classification problem of SVL-MNIST more challenging. Otherwise, all models would obtain high accuracy. This would have made it hard to see the effect of each individual design decision.  We also chose the WineReview dataset because classifying wines based on written text alone is particularly challenging, as evidenced by public leaderboards.
>
> **New results**. The table below shows the results of our new experiments. Both GPT and LoReTTa were trained on disjoint (video, audio) and (video, text) pairs to solve the problem of cross-modal translation. We then evaluated the models on the unseen task of audio captioning. As can be seen, LoReTTa is more than capable of overcoming the modality gap and rivals the upper-bound models (MMGPT). More details can be found in the global rebuttal.
>
> | Method $\phantom{.}$ | $\phantom{...}$ Train | $\phantom{a}$ Test | BLEU4 | METEOR | ROUGE |
> |----------|----------|----------|----------|----------|----------|
> | GPT | A $\rightarrow$ V, V $\rightarrow$ T | $\phantom{.}$ A $\rightarrow$ T |  $\phantom{...}$ 1.7 |  $\phantom{...}$ 18.5 | $\phantom{...}$  30.7 |
> | LoReTTa | $\phantom{.}$ A $\leftrightarrow$ V $\leftrightarrow$ T | $\phantom{.}$ A $\rightarrow$ T | $\phantom{...}$ 2.8 |  $\phantom{...}$ 20.8 |  $\phantom{...}$ 34.7 |
> |     |     |     |     |
> | MMGPT | $\phantom{...}$ A $\rightarrow$ T | $\phantom{.}$ A $\rightarrow$ T | $\phantom{...}$ 6.7 | $\phantom{...}$ 19.4 | $\phantom{...}$ 27.1 |
> | MMGPT| $\phantom{...}$ V $\rightarrow$ T |$\phantom{.}$  V $\rightarrow$ T | $\phantom{...}$ 7.8 | $\phantom{...}$  21.3 | $\phantom{...}$ 29.1 |
>
>
> **References**
>
> Chen et al., Generative Pretraining from Pixels, 2020
>
> Hayes et al., MUGEN: A Playground for Video-Audio-Text Multimodal Understanding and GENeration, 2022
>
> Jaegle et al., Perceiver: General Perception with Iterative Attention, 2021
>
> Yu et al., Scaling Autoregressive Models for Content-Rich Text-to-Image Generation, 2022

---

> > ### Comment · Reviewer_SR2A · 2023-08-11
> > **Follow-up**
> >
> > I appreciate the author responses. I still have a few follow-up comments.
> >
> > **1. I still don't understand why not include the suggested upper bound setups. These should be provided as a reference because the datasets you used contain all 3 modalities for each sample.**
> >
> > **2. Even a 1-layer linear NN achieves 88% test accuracy on MNIST (as can be found in Yann Lecun's original MNIST database page) and a simple KNN can achieve 95% test accuracy.**
> >
> > **3. The authors did not answer my question on the poor accuracy on the text modality of SVL-MNIST dataset.**

---

> > > ### Author Response · Authors · 2023-08-11
> > >
> > > We would like to answer all of the Reviewer's remaining open questions:
> > >
> > > **(1)** As explained in the rebuttal, we did not add the upper bound results due to computational constraints. During the rebuttal, we committed all resources to provide the large-scale experiment. We are currently running the upper bound experiments for TCGA-OMICS and MNIST-SVL, which the Reviewer correctly points out is an interesting comparison (for the MUGEN dataset, the upper bound is given and LoReTTa manages to come close or even exceed it). We will include the results in the final manuscript. If the results are available before the end of the discussion period, we will inform the Reviewer here. In particular, we pre-train two new models and fine-tune them a total of 28 times.
> > >
> > > **(2)** Our experiments on MNIST differ from the results mentioned by the Reviewer because we considered a few-shot scenario (100 samples per class). If we had probed our linear classifier with all classes, we would have gotten well over 90% accuracy. This would have made it difficult to see the impact of the powerful multimodal features learned by LoReTTa, which are useful for downstream tasks with few labeled samples, as can be seen in our experiments. For reference, when we fine-tune on all labels, we get 99.6% accuracy for image, 96.3% for audio, and 82.0% for text.
> > >
> > > **(3)** We have answered this question in the rebuttal (linear probing accuracy, last two sentences). The text modality from SVL-MNIST is exactly the WineReviews dataset. It is indeed noisy, since many words used to describe one wine could be attributed to another wine. And since taste is a subjective experience, two wine tasters could give different reviews for the same wine. We chose this dataset precisely to show the ambiguity of relying on only one modality.

---

> > > ### Author Response · Authors · 2023-08-14
> > > **Upper Bound Results**
> > >
> > > We would like to update the reviewer about the "upper bound experiments". It is trained with C2M3 on all aligned modalities. We call it "C2M3 (3-modal)" and not "Upper Bound Model" because training with all modalities does not guarantee the best result due to negative transfer and modality competition, which can be mitigated by more advanced techniques like transitive modeling as shown by LoReTTa.
> > >
> > > For SVL-MNIST, we have highlighted the accuracies of the modalities that were not seen during pre-training. The subscript indicates the linking modality, i.e., image (I), text (T), and speech (W). For the TCGA-OMICS experiments, we have highlighted the best c-index without the "upper bound". As can be seen, LoReTTa is consistently close to or even better than the full model. These results are also consistent with those from the large-scale experiment on the MUGEN dataset.
> > >
> > >
> > > Method | IMG | TXT | WAV | IMG-TXT | IMG-WAV | TXT-WAV | IMG-TXT-WAV
> > > |----------|----------|----------|----------|----------|----------|----------|----------|
> > > LoReTTa$_I$ | 82.7 | 62.8 | 82.5 | $\phantom{..}$ 88.5 | $\phantom{..}$ 89.7 | $\phantom{..}$ $\textbf{84.0}$ | $\phantom{....}$ $\textbf{90.7}$ |
> > > LoReTTa$_T$ | 81.2 | 63.3 | 81.0 | $\phantom{..}$ 89.0 | $\phantom{..}$ $\textbf{80.9}$ | $\phantom{..}$ 85.5  | $\phantom{....}$ $\textbf{87.8}$ |
> > > LoReTTa$_W$ | 80.1 | 62.1 | 84.2 | $\phantom{..}$ $\textbf{83.0}$ | $\phantom{..}$ 90.4 | $\phantom{..}$ 89.0 | $\phantom{....}$ $\textbf{91.6}$ |
> > > | | | | | | | | |
> > > C2M3 (3-modal)  | 80.8 | 56.6 | 82.4 | $\phantom{..}$ 84.9 | $\phantom{..}$ 86.3 | $\phantom{..}$ 84.9 | $\phantom{....}$ 88.3 |
> > > | | | | | | | | |
> > >
> > > Method | mRNA | miRNA | RPPA | mRNA-miRNA | mRNA-RPPA | miRNA-RPPA | mRNA-miRNA-RPPA
> > > |----------|----------|----------|----------|----------|----------|----------|----------|
> > > BERT | 0.592 | 0.621 | 0.594 |  $\phantom{....}$ 0.595 | $\phantom{....}$ 0.613 | $\phantom{....}$ 0.594 | $\phantom{........}$ 0.610 |
> > > GPT | 0.575 | 0.590 | 0.611 |  $\phantom{....}$ 0.585 | $\phantom{....}$ 0.576 | $\phantom{....}$ 0.606 | $\phantom{........}$ 0.588 |
> > > CLIP | 0.561 | 0.603 | 0.587 |  $\phantom{....}$ 0.610 | $\phantom{....}$ 0.600 | $\phantom{....}$ $\textbf{0.623}$ | $\phantom{........}$ 0.612 |
> > > C2M3 | 0.621 | 0.599 | 0.599 |  $\phantom{....}$ 0.624 | $\phantom{....}$ 0.620 | $\phantom{....}$ 0.571 | $\phantom{........}$ 0.624 |
> > > LoReTTa | $\textbf{0.652}$ | $\textbf{0.623}$ | 0.563 |  $\phantom{....}$ $\textbf{0.660}$ | $\phantom{....}$ $\textbf{0.652}$ | $\phantom{....}$ $\textbf{0.623}$ | $\phantom{........}$ $\textbf{0.657}$ |
> > > | | | | | | | | |
> > > C2M3 (3-modal) | 0.665 | 0.635 | 0.645 | $\phantom{....}$ 0.643 | $\phantom{....}$ 0.671 | $\phantom{....}$ 0.650 | $\phantom{........}$ 0.620 |
> > > | | | | | | | | |

---

> > > > ### Comment · Reviewer_SR2A · 2023-08-17
> > > > **Thank you for the additional experiments and clarification**
> > > >
> > > > Thank you for the additional experiments and clarification.
> > > >
> > > > I have one final doubt about hyperparameter selection. From both the main paper and supplemental, it seems all experiments were using the same set of hyperparameters as the author stated *"We have found that this set of hyperparameters works very well for all experiments"* (Line 269). I believe **the author should report how you perform grid search of the hyperparameters using the validation set**. For example, you should list the range of learning rate and weight decay you have tried on each of your baseline methods, including CLIP and GPT.
> > > >
> > > > Also, will the code and dataset be made public?

---

> > > > > ### Author Response · Authors · 2023-08-18
> > > > >
> > > > > We appreciate the reviewer's comment and the opportunity to clarify. Please allow us to address the last remaining question: Finding hyperparameters for pre-training transformers is very time consuming and resource intensive. We follow best practices developed in our community for implementing state-of-the-art speech, vision, and audio models. In particular, we use a combination of heuristics, theory, and grid search to find the best set of values for learning rate, weight decay, and batch size [Brown et al., 2020; Baevski et al., 2023; Copet et al., 2023; Chowdhery et al., 2023; Hoffmann et al., 2022; Ramesh et al., 2021; Touvron et al., 2023; Yu et al., 2023].
> > > > >
> > > > > Analysis of the training details of recent transformer-based foundation models [see references below and related work section] suggests that a good range for the learning rate and weight decay lies in the interval [1e-3, 1e-4] and [1e-1, 1e-2]. Thus, we searched the {1e-1, 1e-2, 1e-3, 1e-4, 1e-5, 1e-6} x {0e-0, 1e-1, 1e-2, 1e-3} grid to confirm this range. We then searched the refined {6e-4, 3e-4, 1e-4} x {1e-1, 5e-2, 1e-2} grid to find the final hyperparameters. The final learning rate of 6e-4 and weight decay of 1e-1 works for all models that use some kind of causal or masked modeling (BERT, GPT, C2M3, LoReTTa) similar to Artetxe et al. (2023).
> > > > >
> > > > > This is consistent with the hyperparameters used in the reported literature. Since end-to-end training would be too expensive, we performed the search with a smaller model. Recent research shows that many optimal hyperparameters remain stable even when the model size changes [Yang et al. 2021]. For the large-scale experiment on the MUGEN dataset, we used the same hyperparameters as the upper bound MMGPT model, which is also in the above range of optimal hyperparameters. More about hyperparameter stability can be found in Kaplan et al. (2020) and McCandlish et al. (2018).
> > > > >
> > > > > Now, in the paper and the appendix, we stated that CLIP [Radford et al., 2021] was trained with a different set of hyperparameters. Contrastive pre-training is very different from all previous methods, which are based on ideas from natural language processing.  We used CLIP and models based on CLIP [e.g., Girdhar et al., 2023] as an anchor to find the learning rate, weight decay, and temperature using heuristics and simple grid search similar to those described above.
> > > > >
> > > > > As stated in the manuscript, all datasets used are public. We will release code for downloading and preprocessing the dataset to obtain the aligned modalities. The architecture of the transformer is well known; and the specific model details used for SVL-MNIST as well as TCGA-OMICS are described in the paper. For the MUGEN experiment, we utilize the same transformer model as in the reference paper [Hayes et al., 2022]. Moreover, our training can be easily reproduced with the extensive methodological details provided in the submission.
> > > > >
> > > > > **References**
> > > > >
> > > > > Artetxe et al., On the Role of Bidirectionality in Language Model Pre-Training, 2022
> > > > >
> > > > > Brown et al. Language models are few-shot learners, 2020
> > > > >
> > > > > Baevski et al., Efficient Self-supervised Learning with Contextualized Target Representations for Vision, Speech and Language, 2023
> > > > >
> > > > > Copet et al., Simple and Controllable Music Generation, 2023
> > > > >
> > > > > Chowdhery et al., PaLM: Scaling Language Modeling with Pathways, 2023
> > > > >
> > > > > Girdhar et al., ImageBind: One Embedding Space To Bind Them All, 2023
> > > > >
> > > > > Hayes et al., MUGEN: A Playground for Video-Audio-Text Multimodal Understanding and GENeration, 2022
> > > > >
> > > > > Hoffmann et al., Training Compute-Optimal Large Language Models, 2022
> > > > >
> > > > > Kaplan et al., Scaling Laws for Neural Language Models, 2020
> > > > >
> > > > > McCandlish et al., An Empirical Model of Large-Batch Training, 2018
> > > > >
> > > > > Radford et al., Learning Transferable Visual Models From Natural Language Supervision, 2021
> > > > >
> > > > > Ramesh et al., Zero-Shot Text-to-Image Generation, 2021
> > > > >
> > > > > Touvron et al., Llama 2: Open Foundation and Fine-Tuned Chat Models, 2023
> > > > >
> > > > > Yang et al., 2022, Tuning large neural networks via zero-shot hyperparameter transfer, 2021
> > > > >
> > > > > Yu et al., Scaling Autoregressive Multi-Modal Models: Pre-Training and Instruction Tuning, 2023

---

> > > > > > ### Comment · Reviewer_SR2A · 2023-08-18
> > > > > > **Thanks for the clarification**
> > > > > >
> > > > > > Thank you for the clarification and attaching the grid of hyperparameters you used for all experiments besides CLIP.
> > > > > >
> > > > > > Could you also attach the grid of hyperparameters you used for CLIP?
> > > > > >
> > > > > > I am confused by your suggested reference on transferring hyperparameters from smaller models to larger models. In particular, you referenced [Yang et al. 2021], but that paper initialized the model using a special technique called "Maximal Update Parametrization" which was never mentioned in your manuscript. Without this technique, I don't see how the hyperparameters can transfer across model sizes.
> > > > > >
> > > > > > I also agree that CLIP is very different from the rest of the baselines. CLIP is also very sensitive to one additional hyperparameter you did not seem to tune, which is **batch size**. From your supplemental Table A2, it seems you used a **rather small batch size of 16**. Recent work such as [1] already scaled up CLIP's batch size from 8k to 16k and show further improvements. Even the work you cited for [Girdhar et al., 2023] *"as an anchor"* used a minimal batch size of 512 in all of their experiments. I understand that you have computational constraint, but because CLIP needs a large batch size to mine for hard negative samples in the current batch, using a smaller batch size defeats this purpose and will render suboptimal results. The authors should clearly acknowledge this in the paper.
> > > > > >
> > > > > >
> > > > > > **References:**
> > > > > >
> > > > > > [1] Image Captioners Are Scalable Vision Learners Too. Tschannen et al. 2023

---

> > > > > > > ### Author Response · Authors · 2023-08-18
> > > > > > >
> > > > > > > Dear Reviewer,
> > > > > > >
> > > > > > > We would like to clarify the confusion caused by the previous comment. We did not transfer the hyperparameters 1-to-1 from a smaller model to a larger model. As shown in Yang et al. (2021) and the other referenced papers, one does indeed need to adjust the hyperparameters for different model sizes. However, a look at Figure 1 in [Yang et al. 2021] and the implementation of other similar models shows that the optimal learning rate is often in the range [1e-3, 1e-5]. Since very small learning rates are mostly needed to train very large models, the range [1e-3, 1e-4] is reasonable and also standard in the literature. This is what we meant by stability. The confusion arose because Yang et al. (2021) use the word stability in the context of their proposed zero-shot transfer of hyperparameters.
> > > > > > >
> > > > > > > As a sanity check, we confirmed the above range by searching the coarse grid with a smaller model. For the finer grid, we then used our final model. To reassure the reviewer that we are not using the same hyperparameters for all model sizes, we would like to point out that the learning rate for the 512-width model on the SVL-MNIST and TCGA-OMICS datasets is 6e-4, and for the 768-width model on the MUGEN dataset is 3e-4 (we will also include the hyperparameters for the new experiments in the final manuscript).
> > > > > > >
> > > > > > > The learning rate and weight decay grid for our CLIP experiments are {1e-3, 5e-4, 1e-4, 5e-4, 1e-5} x {0.5, 0.2, 0.1, 0.05}. The range of values is derived from CLIP and similar work. We agree with the reviewer that a drawback of contrastive training is the need for very large batch sizes. This is very difficult to achieve without huge GPU clusters. We thank the reviewer for pointing this out and we will clearly acknowledge the potential implication in the final paper.

---

### Official Review · Reviewer_FSDR · 2023-07-06

**Soundness:** 2 fair
**Presentation:** 3 good
**Contribution:** 2 fair
**Rating:** 6
**Confidence:** 2

**Summary:**

This paper proposes a new method LoReTTa (Linking mOdalities with a tRansitive and commutativE pre-Training sTrAtegy) to address the setting where not all modality pairs are available during training and inference. Concretely, it leverages casual masked modeling and transitive modeling to accomplish the objective. Theoretical analysis has been provided to help the understanding of the proposed method, with empirical results presented as well.

**Strengths:**

+ The target problem is practical and meaningful, which may indeed occur in our real implementation of multimodal algorithms.

+ The proposed method is simple and easy-to-follow.

+ Theoretical analysis contributes to the understanding.

**Weaknesses:**

I am not an expert in this field, but I find some issues that might affect the quality of this manuscript.

-  The proposed method seems a bit tricky; I am concerned whether this is up to the standard of NeurIPS. And the authors should implement ablation studies to verify the effectiveness of each proposed technique.

- In transitive modeling, we need to use a "predicted" modality to bridge the missing modality pair. However, since the prediction can not be that accurate, how can we ensure robustness against such noise?

- The experimental datasets are sort of small (e.g. SVL-MNIST dataset). It would be better if experiments are conducted on more datasets or even larger ones.



**Questions:**

Please refer to the weaknesses.

**Limitations:**

None.

---

> ### Author Rebuttal · Authors · 2023-08-07
>
> **Dear Reviewer FSDR,**
>
> It is a great pleasure to hear that our work addresses a practical and meaningful problem. Our proposed algorithm is indeed simple but has a big impact. We demonstrated this by applying our novel method to a real medical problem. The results we saw were fully consistent with our theoretical analysis of the mechanisms behind LoReTTa. In the following sections, we would like to address the remaining concerns.
>
> **Ablation studies.** We have not forgotten about the ablation studies. They are critical to understanding which component contributes to which outcome. The two new concepts we have introduced are commutative modeling and transitive modeling. We applied commutative modeling to CM3 [Aghajanyan et al., 2022] to get C2M3. We then extended C2M3 with transitive modeling – resulting in LoReTTa. CM3 and its variants [Aghajanyan et al., 2022; Bavarian et al., 2022; Fried et al., 2023] are themselves extensions of GPT [Radford et al., 2018] and BERT [Devlin et al., 2018]. Thus, an ablation study for causal masked modeling has already been done in the literature (they show noticeable improvements).  Nevertheless, we used all of these models as baselines in our real-world experiment on TCGA-OMICS, comparing GPT, BERT, C2M3 (GPT + BERT + Commutative Modeling), LoReTTa (C2M3 + Transitive Modeling), as well as CLIP  [Radford et al., 2022]. Hence, our benchmark automatically included an ablation study. Since C2M3 is already a strong improvement over GPT, BERT, and CLIP, we only compared LoReTTa with C2M3 in our SVL-MNIST experiments, where we extensively studied the effect of transitive modeling under different scenarios.
>
> **Robustness against noise.** LoReTTa predicts the missing modality to bridge the modality gap. This only works if the prediction is accurate. Otherwise, we would be training with noisy samples. We ensure this by pre-training our transformer with C2M3 until convergence. It learns to effectively transition between all seen modality pairs through causal modeling, within modalities through masked modeling, and vice versa through commutative modeling. With this strong generative model, we are able to faithfully infer the missing modality. We further reduce the noise by adding another level of alignment: using the predicted modality to infer the omitted sample (see Figure 2 for more details).
>
> **Large-scale experiments.** We chose SVL-MNIST to pre-train 12 models and fine-tune them a total of 162 times. This allowed us to gain valuable insights into the inner workings of transitive modeling with a limited computational budget. As a real-world application, we chose the medical domain, which is known to lack large datasets for training foundation models. It is still worth noting that LoReTTa already gives impressive results on a smaller medical dataset with highly complex modalities such as genomics, transcriptomics, or proteomics. Nevertheless, it is interesting to see how well LoReTTa scales to larger experiments. Thanks to the valuable feedback we received during the rebuttal, we were directed to a new multimodal dataset with 375,000 samples from a real-world vision-language-audio problem [Hayes et al., 2022]. We immediately downloaded the data and started training. The results are presented in the global rebuttal and show that LoReTTa is scalable and effective in more complex (non-medical) environments.
>
> **New results**. The table below shows the results of our new experiments. Both GPT and LoReTTa were trained on disjoint (video, audio) and (video, text) pairs to solve the problem of cross-modal translation. We then evaluated the models on the unseen task of audio captioning. As can be seen, LoReTTa is more than capable of overcoming the modality gap and rivals the upper-bound models (MMGPT). More details can be found in the global rebuttal.
>
> | Method $\phantom{.}$ | $\phantom{...}$ Train | $\phantom{a}$ Test | BLEU4 | METEOR | ROUGE |
> |----------|----------|----------|----------|----------|----------|
> | GPT | A $\rightarrow$ V, V $\rightarrow$ T | $\phantom{.}$ A $\rightarrow$ T |  $\phantom{...}$ 1.7 |  $\phantom{...}$ 18.5 | $\phantom{...}$  30.7 |
> | LoReTTa | $\phantom{.}$ A $\leftrightarrow$ V $\leftrightarrow$ T | $\phantom{.}$ A $\rightarrow$ T | $\phantom{...}$ 2.8 |  $\phantom{...}$ 20.8 |  $\phantom{...}$ 34.7 |
> |     |     |     |     |
> | MMGPT | $\phantom{...}$ A $\rightarrow$ T | $\phantom{.}$ A $\rightarrow$ T | $\phantom{...}$ 6.7 | $\phantom{...}$ 19.4 | $\phantom{...}$ 27.1 |
> | MMGPT| $\phantom{...}$ V $\rightarrow$ T |$\phantom{.}$  V $\rightarrow$ T | $\phantom{...}$ 7.8 | $\phantom{...}$  21.3 | $\phantom{...}$ 29.1 |
>
>
> **References**
>
> Aghajanyan et al., CM3: A Causal Masked Multimodal Model of the Internet, 2022
>
> Bavarian et al., Efficient Training of Language Models to Fill in the Middle, 2022
>
> Devlin et al., BERT: Pre-training of Deep Bidirectional Transformers for Language Understanding, 2018
>
> Fried et al, InCoder: A Generative Model for Code Infilling and Synthesis, 2023
>
> Hayes et al., MUGEN: A Playground for Video-Audio-Text Multimodal Understanding and GENeration, 2022
>
> Radford et al., Improving Language Understanding by Generative Pre-Training, 2018
>
> Radford et al., Learning Transferable Visual Models From Natural Language Supervision, 2022

---

### Official Review · Reviewer_7Goz · 2023-07-08

**Soundness:** 4 excellent
**Presentation:** 2 fair
**Contribution:** 3 good
**Rating:** 7
**Confidence:** 3

**Summary:**

This paper aims to remedy the problem of training a mobility to perform well on any combination of modalities in the training data, regardless of if these combinations appear in the training data. This is done through commutative and transitive pretraining: the former allows for the model to learn modality A from B and B from A, and the latter learns the missing joint distributions. This results in a model which outperforms models trained on fewer modalities in the pre-training stage or models trained without transitive modeling.

**Strengths:**

This paper addresses an important and under-addressed problem of training with mismatching modality pairings, and their commutative and transitive modeling is proven to be very affective on training in both toy and real datasets. While I do not keep up closely with the literature in many-modality pretraining, if this is the first paper to address this problem of training on mismatching modalities, it is incredibly valuable to the community and would be a good baseline for any future methods due to its simplicity.

**Weaknesses:**

My main concerns are surrounding the writing, as I found the methods section to be quite wordy and I think you could remove a lot of pretext on transformers and existing pretraining strategies. Furthermore, possibly due to the density of the method section, I found the explanation of the perplexity to be a bit confusing.

Aside from the writing, I wonder what the difference in compute requirements is between the authors method and the baselines. It was mentioned that LoReTTa was trained on on A100 for 3 days, which seems like a large amount of compute for the datasets presented.

**Questions:**

see above

**Limitations:**

The main limitation I think is missing is the amount of compute required.

---

> ### Author Rebuttal · Authors · 2023-08-07
>
> **Dear Reviewer 7Goz,**
>
> We sincerely appreciate your critical insights and expertise during the review of our manuscript. We are greatly encouraged by your recognition that our work addresses an important and understudied problem. Such recognition underscores the relevance of our research and further motivates us. Below, we would like to address your two concerns in detail.
>
> **Writing.** Our proposed learning algorithm covers a wide area of research including transformers, computer vision, natural language processing, audio signal processing, computational pathology, self-supervised learning, and multimodal learning. We sought to appropriately contextualize our method within the existing literature and bring readers from a wide variety of backgrounds on the same page, as we believe that LoReTTa will have a profound impact in a wide variety of fields. Still, we appreciate your concern and will make the related work and methods section even more precise.
>
> **Compute requirements.** LoReTTa relies on the auto-regressive generation of the missing modality for transitive training. This is a slow process for transformer decoders. However, we are happy to report that our community has made great progress in reducing computational complexity. Recently, FlashAttention-2 [Dao, 2023] has been announced, which doubles the speed of FlashAttention [Dao et al., 2022]. Multi-query attention [Shazeer, 2019] and grouped-query attention [Ainslie et al., 2023] have been shown to reduce inference time by a factor of up to 6. In addition, speculative sampling [Chen et al., 2023] achieves a 2-2.5x decoding speedup. The combination of all these advances promises to significantly reduce the training time of LoReTTa. We have omitted them to avoid too many moving parts in our experiments that would distract from the main results. In the global rebuttal, we presented the results of an additional experiment on a large-scale dataset [Hayes et al., 2022] with 375,000 samples. We were able to achieve excellent results with LoReTTa with a limited time budget of just a few days.
>
> **New results**. The table below shows the results of our new experiments. Both GPT and LoReTTa were trained on disjoint (video, audio) and (video, text) pairs to solve the problem of cross-modal translation. We then evaluated the models on the unseen task of audio captioning. As can be seen, LoReTTa is more than capable of overcoming the modality gap and rivals the upper-bound models (MMGPT). More details can be found in the global rebuttal.
>
> | Method $\phantom{.}$ | $\phantom{...}$ Train | $\phantom{a}$ Test | BLEU4 | METEOR | ROUGE |
> |----------|----------|----------|----------|----------|----------|
> | GPT | A $\rightarrow$ V, V $\rightarrow$ T | $\phantom{.}$ A $\rightarrow$ T |  $\phantom{...}$ 1.7 |  $\phantom{...}$ 18.5 | $\phantom{...}$  30.7 |
> | LoReTTa | $\phantom{.}$ A $\leftrightarrow$ V $\leftrightarrow$ T | $\phantom{.}$ A $\rightarrow$ T | $\phantom{...}$ 2.8 |  $\phantom{...}$ 20.8 |  $\phantom{...}$ 34.7 |
> |     |     |     |     |
> | MMGPT | $\phantom{...}$ A $\rightarrow$ T | $\phantom{.}$ A $\rightarrow$ T | $\phantom{...}$ 6.7 | $\phantom{...}$ 19.4 | $\phantom{...}$ 27.1 |
> | MMGPT| $\phantom{...}$ V $\rightarrow$ T |$\phantom{.}$  V $\rightarrow$ T | $\phantom{...}$ 7.8 | $\phantom{...}$  21.3 | $\phantom{...}$ 29.1 |
>
>
> **References**
>
> Ainslie et at., GQA: Training Generalized Multi-Query Transformer Models from Multi-Head Checkpoints, 2023
>
> Chen et al., 2023 Accelerating Large Language Model Decoding with Speculative Sampling, 2023
>
> Dao et al., FlashAttention: Fast and Memory-Efficient Exact Attention with IO-Awareness, 2022
>
> Dao, FlashAttention-2: Faster Attention with Better Parallelism and Work Partitioning, 2023
>
> Hayes et al., MUGEN: A Playground for Video-Audio-Text Multimodal Understanding and GENeration, 2022
>
> Shazeer, Fast Transformer Decoding: One Write-Head is All You Need, 2019

---

> > ### Comment · Reviewer_7Goz · 2023-08-18
> > **Response**
> >
> > Thank you for the detailed response. After reviewing the comments of the other reviewers, my thoughts on the utility of the paper has not changed but I do have similar concerns to QEx4 about reproducibility and paper structure, with most of my concerns on the former. I am unsure why the authors said they would release code to process the data but not the full training code and this sparks a fair amount of concern. Even with a detailed description of the implementation, not releasing the code to reproduce the experiments exactly leaves plausible deniability if others implementations do not perform similarly.
> >
> > I apologize for responding so late and I don't expect and further experimental results or detailed explanations in the remaining response time. That being said, the concerns around reproducibly have cause me to reduce my score from a strong accept to accept, but I am happy to hear from the authors if they think my concerns are not well founded.

---

> > > ### Author Response · Authors · 2023-08-18
> > >
> > > We fully understand the reviewer's concerns and agree that reproducibility is an important topic. At this time, we are unable to make our code publicly available. This decision is beyond the direct control of the authors. Given our constraints, we have tried to provide as much detail as possible in the paper, appendix, and rebuttal/discussion period to help readers implement our proposed method. We are working behind the scenes to publish the code, but cannot promise a release.

---

> > > ### Author Response · Authors · 2023-08-20
> > > **Pseudo Code**
> > >
> > > In addition to the details outlined in the paper, we will also include a pseudocode in the final submission to better illustrate our approach:
> > >
> > > ```
> > > class LoReTTa:
> > >     """
> > >     Pseudo-code for commutative and transitive modeling
> > >     """
> > >     def forward(self, tokens, modes=['commutative','transitive']):
> > >         """
> > >         tokens ... tokenized inputs, e.g., [x_0,...x_n, y_0,...,y_m]
> > >         x_0, y_0 .... modality-specific tokens, 'a', 'b', or 'c'
> > >         """
> > >
> > >         if 'commutative' in modes: #shuffle modalities
> > >             tokens = self.shuffle_modalities(tokens)
> > >
> > >         if 'transitive' in modes: #generate missing modality
> > >             existing_modalities = self.extract_modality_tokens(tokens)
> > >
> > >             if ['a', 'b'] in existing_modalities: #case 1
> > >                 modality_a, modality_b = self.split_tokens(tokens)
> > >                 modality_c = self.model.generate([modality_b, 'c'])
> > >                 tokens = [modality_c, modality_a]
> > >
> > >             if ['b', 'c'] in existing_modalities: #case 2
> > >                 modality_b, modality_c = self.split_tokens(tokens)
> > >                 modality_a = self.model.generate([modality_b, 'a'])
> > >                 tokens = [modality_a, modality_c]
> > >
> > >         logits = self.model(tokens[:, :-1]) #get predictions
> > >         targets = tokens[:, +1:] #shift targets
> > >
> > >         loss = self.criterion(logits, targets) #calculate cce-loss
> > >         return self.split_loss(loss) #return individual loss for each modality
> > > ```

---

### Official Review · Reviewer_QEx4 · 2023-07-12

**Soundness:** 2 fair
**Presentation:** 4 excellent
**Contribution:** 3 good
**Rating:** 6
**Confidence:** 4

**Summary:**

In this paper, the authors introduce the problem of learning from multi-modal data (e.g. modalities A, B, and C) if not all modality combinations are available at training time (e.g. only paired data (A, B) and (B, C) are given). This constitutes a highly relevant research problem, as learning from multiple modalities has shown a lot of potential, but collecting fully annotated multi-modal datasets is a costly endeavour.

In order to take advantage of _existing_ data and modality combinations, the authors propose a self-supervised learning paradigm (LoReTTa), which combines cycle consistency with masked modelling: i.e., the models are trained to predict masked tokens in A from B (masked modelling) or C from A via B (masked modelling with cycle consistency). As a result, the underlying model is able to perform well not only on the seen modality combinations (A, B) and (B, C), but is also able to handle unseen combinations such as (A, C) or (A, B, C).

In particular, the authors show on a synthetically created data pairing (A=MNISt + B=AudioMNIST + C=WineReviews) that a model trained with cycle consistency (LoReTTa) achieves significantly lower perplexity and higher linear probing accuracy on the test data than models trained without such cycle consistency. Additionally, the authors evaluate their approach on the real-world TCGA-OMICS dataset and report consistent performance improvements, especially on unseen modality combinations.

**Strengths:**

This paper is a good submission for the following reasons.

- S1: The authors introduce a novel and highly relevant setting for learning from multiple modalities.
- S2: The proposed approach is well-motivated and simple yet effective.
- S3: The presentation and writing make this paper a pleasure to read.
- S4: The experimental results show convincing and significant improvements over the chosen baseline models (however, see W1).

**Weaknesses:**

While I find this to be a good submission in general, there are several points of concern that make me hesitant. Specifically, I would highly appreciate additional feedback from the authors on the following aspects.

- W1 (baselines and evaluation): The two main novel contributions of this paper seem to be (1) introducing the problem of learning from multi-modal data with disjoint modality pairings, and (2) showing that cycle consistency can be highly beneficial in this context. In order to validate (2), the authors compare to backbones without this cycle consistency in Tables 1+2, and to other training paradigms (contrastive loss, CLIP) or that use different model architectures without cycle consistency (different attention masks in GPT and BERT) in Table 3. While C2M3 seems to be a strong contender among the models without cycle consistency (Table 3), the other baselines yield competitive results, too. __My question in this context is thus the following__: why do the authors only include the cycle consistency in the C2M3 model? In principle, cycle consistency could also be integrated into BERT and GPT. It seems to me like the submission could be strengthened by showing that all of the baselines benefit from the addition of cycle consistency.
- W2 (causal modelling): Following from W1: what is the motivation to focus on causal masked modelling in the first place? In the context of multi-modal learning, I fail to see why a causal attention structure is desired. Even more, only due to the causal modelling approach does the need for commutative switching arise, which the authors introduce as an important contribution. If the authors were to apply the cycle consistency approach to BERT e.g., this would not be necessary.
- W3 (structure of the paper): Especially in light of W1 and W2, the space usage in the paper seems to be suboptimal to me. E.g., half a page in the method section is spent on the model architecture and tokenization options, which seem to be experimental details rather than essential aspects of the proposed method. Similarly, the definition and discussion of what constitutes a modality seems to be tangential, yet is given another 1/2 page in section 4. More generally, while interesting to read, the relevance of section 4 and its integration with the rest of the paper are not fully convincing to me. Instead, I think the paper would benefit from an extended experimental evaluation, including additional comparisons to other methods under the addition of cycle consistency (see W1) and potentially additional datasets and methods (e.g., as in [4]), which would allow for a more fine-grained understanding of the benefits and limitations of the proposed method (e.g., how much does it help if full modality combinations are in fact, at least sometimes, available?)
- W4 (reproducibility), following from W3: While the authors spend a significant amount of space on describing tokenization strategies, it remains unclear to me when and how e.g. VQ-VAEs or CNNs are used, as the authors only mention the usage of PCA in the experimental details. With the current amount of information given, I am concerned about the reproducibility, as the final model architectures and the tokenization (where do CNNs and VQ-VAEs enter the picture) remain unclear to me.



Additional minor remarks:
- In the light of the strong results reported in ImageBind [29], it would be interesting to include additional discussion on how LoReTTa conceptually compares to CLIP (bringing 'transitivity' to masked modelling, which is somewhat inherent in CLIP?), how LoReTTa might perform in such a large scale setting, and why CLIP might perform suboptimally in the reported results (Table 3). Does CLIP simply require more data?
- The authors argue that their transitivity approach is 'much more general' than cycle consistency (see caption Fig. 2, or line 153). I currently fail to see the difference, however, and would appreciate if the authors could elaborate.
- Line 313: 'communicative switching' -> do the authors mean commutative?

**Questions:**

Please see the weakness section. If the authors are able to address my concerns, I will gladly update my score.

**Limitations:**

The authors have discussed societal impacts and limitations of their work. Nonetheless, as discussed in the weakness section, I believe the paper could further benefit from an increased discussion and comparison to related works.

---

> ### Author Rebuttal · Authors · 2023-08-07
>
> **Dear Reviewer QEx4,**
>
> We are very grateful for the thorough review and appreciate that our proposed method is considered novel and highly relevant, showing significant and consistent improvements across different baselines. While we have addressed many of the Reviewer's concerns in the global rebuttal, we provide a customized response below.
>
> **W1 (baselines and evaluation).** We refer to our approach as transitive modeling instead of cycle consistency [Zhu et al., 2017] because, upon closer inspection, the two are quite different. CycleGAN was proposed to transition between two domains (image styles) of the same modality (image), while LoReTTa can transition between three or more modalities (i.e., image, text, and audio). It also avoids using the input as a target, which could cause the representation to collapse. Furthermore, our approach works with more complex modality relations, as described in the global rebuttal. All this makes transitive modeling much more general and applicable than cycle consistency – with which it again shares only some high-level conceptual similarities.  By design of transitive modeling, the missing modality must be generated. This is not possible with BERT [Devlin et al., 2018]. It is trained with masked modeling, which simply predicts the missing tokens, not a completely missing sequence on the right. On the other hand, GPT-style [Radford et al., 2018] models are generative and can be extended with our proposed transitive modeling approach.  In particular, we start with CM3, which improves GPT by introducing causal masked modeling [Aghajanyan et al., 2022; Bavarian et al., 2022; Fried et al., 2023]. This effectively combines BERT and GPT into a single model. We extend CM3 with commutative (C2M3) and transitive (LoReTTa) modeling.  Thus, with LoReTTa we have a unified architecture that includes transitive modeling in both GPT and also BERT. The latter would not be possible otherwise, as explained above.
>
> **W2 (causal modeling).** To use transitive modeling, we must predict the missing modality by generating it in its entirety. This requires a generative model, which is only possible with causal training in a sequential model. However, we are aware of the advantages of bidirectional attention in BERT over causal masks in GPT. Therefore, we use a unified version called causal masked modeling to take advantage of both. This is our way to use transitive modeling in BERT since it is not intended to be used for generative tasks.
>
> **W3 (structure of the paper).** We provide an additional large-scale experiment on the MUGEN [Hayes et al., 2022] dataset with 375,000 aligned samples from video, audio, and text samples to make our empirical findings even more comprehensive. In particular, we apply LoReTTa to the task of cross-modal translation (the task is analogous to visual question answering). This generative task is not possible with the masked (BERT) and contrastive (CLIP) approaches. Therefore, we compare it only with GPT and its commutative (C2M3) and transitive version (LoReTTa). Thanks to the Reviewer's suggestions, we will restructure our paper to make room for this additional experiment.
>
> **W4 (reproducibility).** We briefly mentioned our tokenization scheme in the methods and results sections. Since the MNIST dataset contains low-dimensional samples, we encoded each byte stream as a token. For the high dimensional TCGA dataset, we used PCA for dimensionality reduction, and bin each dimension into tokens. For the new dataset, we used pre-trained video and audio VQ-VAE encoders provided along the dataset. We will emphasize this more in the final version.
>
> **Comparison to CLIP and ImageBind.** The advantage of LoReTTa is that it can be used as both an encoder and a decoder to solve both discriminative and generative problems, as demonstrated in our old and new experiments. CLIP [Radford et al., 2022] and its extension ImageBind [Girdhar et al., 2023] (published shortly before the NeurIPS 2023 submission deadline) train their models primarily as encoders – one for each modality. For generative tasks, a diffusion model or other variants must be fitted to these embeddings. Thus, contrastive methods must train multiple encoders and decoders. Worse, contrastive training is inefficient. It requires large batch sizes and large datasets [Chet et al., 2020]. This is because self-supervised contrastive loss only clusters embeddings from the same sample and pushes away those from other samples. Consider a batch with an image of a cat and an accompanying text. In the same batch, there is another image of a similar cat with a similar text description. Now, the CLIP loss will individually push the feature vectors of the first and second image-text pairs closer together. But at the same time, it will push the representations of each sample away from each other, regardless of their similarity.
>
> **New results.** Please have a look at the global rebuttal, where we have added the additional large-scale experiments.
>
>
> **References**
>
> Aghajanyan et al., CM3: A Causal Masked Multimodal Model of the Internet, 2022
>
> Bavarian et al., Efficient Training of Language Models to Fill in the Middle, 2022
>
> Chen et al., A Simple Framework for Contrastive Learning of Visual Representations, 2020
>
> Devlin et al., BERT: Pre-training of Deep Bidirectional Transformers for Language Understanding, 2018
>
> Fried et al, InCoder: A Generative Model for Code Infilling and Synthesis, 2023
>
> Girdhar et al., ImageBind: One Embedding Space To Bind Them All, 2023
>
> Hayes et al., MUGEN: A Playground for Video-Audio-Text Multimodal Understanding and GENeration, 2022
>
> Radford et al., Improving Language Understanding by Generative Pre-Training, 2018
>
> Radford et al., Learning Transferable Visual Models From Natural Language Supervision, 2022
>
> Zhu et al., Unpaired Image-to-Image Translation using Cycle-Consistent Adversarial Networks, 2022

---

> > ### Comment · Reviewer_QEx4 · 2023-08-14
> >
> > I am very grateful for the detailed rebuttal and thank the authors for their effort. For now, I still have the following questions:
> >
> > 1. The authors state that 'This is not possible with BERT [Devlin et al., 2018]. It is trained with masked modeling, which simply predicts the missing tokens, not a completely missing sequence on the right.'. I am not fully convinced by this argument for two reasons, and I would highly appreciate if the authors could clarify a potential misunderstanding from my side. First, predicting 'the missing tokens' versus 'a sequence to the right' seems to differ mainly in the mask layout — I do not understand why instead of a random masking scheme (as done in BERT) one could not simply mask most or the entirety of a given modality (except for modality specific encodings to signal which modality should be reconstructed). This would effectively constitute conditional generation with BERT.  Secondly, the notion of 'a sequence on the right' is confusing to me in the context of transformers, which — without a causal masking structure (as in BERT) — are agnostic to the token order. As I mentioned in W2, BERT would not even require any commutative switching, since the model would be agnostic to the placement of the additional modality. As evidenced by the introduction of the commutative switching, the authors seem to agree with me that a causal dependence between the modalities is not necessarily desirable and BERT would thus lend itself to such a setup.
> >
> > 2. I am still not convinced that the transitive modelling approach is conceptually very different from cycle consistency and I would recommend not running the risk of overclaiming this contribution. The main difference seems to be the extension from a single step ($A\rightarrow B$) to multiple steps ($A\rightarrow B\rightarrow C$); whether or not $A, B, C$ are from the same modality seems to be a semantic detail. In this context, I would also like to point out the 2019 paper [Learning Robust Joint Representations by Cyclic Translations Between Modalities](https://dl.acm.org/doi/10.1609/aaai.v33i01.33016892) as an additional citation that the authors seem to have missed. This paper seems highly relevant to the proposed approach and would kindly like to ask the authors if they could discuss commonalities and differences to this paper within the rebuttal period.

---

> > > ### Author Response · Authors · 2023-08-14
> > >
> > > We appreciate the reviewer's recognition of the time and effort we put into the rebuttal and the additional experiments provided. We would like to address the two remaining questions below.
> > >
> > > **(1.1)** As we understand it, the reviewer's concern is why we did not mask the missing modality and use bidirectional attention for the remaining modality. By reordering the modalities so that all masked modalities are at the end of the input, this proposal is equivalent to prefix modeling as explored in T5 [Raffel et al., 2019] and SimVLM [Wang et al., 2021]. With prefix modeling, the model has the full context of the unmasked modalities while still being able to predict the missing ones. In this regard, we highly recommend the recent paper by Artexte et al. (2022), which explores different pre-training strategies such as causal, masked, and prefix modeling. They show that both causal modeling and causal masked modeling outperform prefix modeling as well as masked modeling for generative tasks. Even for infilling tasks (which is the main learning objective of BERT pre-training), causal modeling and causal masked modeling are still highly competitive compared to masked approaches. Since LoReTTa relies heavily on predicting the missing modality for transitive pre-training, high perplexity in generative tasks is crucial. In fact, we have shown this in our discussion of the error bound. In conclusion, we agree with the reviewer that while it is in principle possible to use BERT for generative tasks, it is not a good foundation for transitive modeling as literature has shown that this yields a weak baseline for the intended use case.
> > >
> > > **(1.2)** A pure transformer is indeed permutation invariant. However, one must add modality-specific positional encoding to correctly capture the spatio-temporal relations of an image, sentence, or song. Thus, the representation of each token depends strongly on its context. In bidirectional masking, the token is allowed to consider the "past" and the "future", whereas in unidirectional masking, a token can only attend to previous information. We agree that the former is equivalent to the latter if all masking happens to be on the right. In practice, this is unlikely. Hence, BERT and GPT are very different models. Again, we refer to the paper by Artexte et al. (2022) for a deeper dive into this very fascinating topic. Their results fit perfectly with what we know about sequence models. Intuitively, a bidirectional mask considers information from both left and right to infer the missing information. It was never designed to rely on left information alone (except in the rare case where only right tokens were randomly masked). There is another good paper about this topic [Wang and Cho, 2019] that explores the generative capability of BERT by iteratively placing masked tokens at the end of the input to generate the output. The results are consistent with the recent paper by Artexte et al. (2022) in that the generated output is inferior to GPT. Since the goal of transitive modeling is to obtain high quality predictions of the missing modality, a causal approach is the best approach, as evidenced in the literature.
> > >
> > > We thank the reviewer for pointing out this topic. It may cause confusion for future readers. That is why, we will include the two mentioned papers in the final version of our paper.
> > >
> > > **(2.0)** Please see the second post for an answer to question number two.

---

> > > > ### Author Response · Authors · 2023-08-14
> > > >
> > > > **(2.0)** We stated in the (global) rebuttal that cycle consistency is conceptually similar to transitive modeling, and we have referenced cycle consistency several times in our paper (notably in the methods section and in Figure 2). We respect previous work that explores cycle consistency and do not hide the similarity. However, despite the similar flavor of both ideas, our proposed method is still different, and we would like to explain this in more detail.
> > > >
> > > > The most popular version of cycle consistency was introduced in CycleGAN. Given an input $a$ in domain / modality $A$, one wants to generate an aligned output $b$ in domain / modality $B$. The model then computes a discriminative loss $D$ on $b$ and a reconstruction loss $L$ on the original $a$ and predicted $\hat{a}$. Another version of cycle consistency was proposed in the recommended paper about MCTN [Pham et al., 2019]. We will discuss the model in more detail below, but here is a brief summary: Given aligned inputs $(a, b)$ from modalities $A$ and $B$, $a$ is used to predict $\hat{b}$, and the predicted $\hat{b}$ is used to predict $\hat{a}$. Then the reconstruction loss $L$ is applied to compare $a$ and $\hat{a}$ as well as $b$ and $\hat{b}$. LoReTTa, on the other hand, starts with aligned modalities $(a, b)$ and $(b', c')$, it uses $b$ to predict $\hat{c}$ and $\hat{c}$ to predict $\hat{a}$, the reconstruction loss then compares $a$ and $\hat{a}$. Since we use commutative modeling, the other direction starting from another sample $(b', c')$ is also learned. In summary, we have the following approaches:
> > > >
> > > > **CycleGAN**: Given $a$, model $a  \rightarrow \hat{b}  \rightarrow \hat{a}$, and calculate $L(a, \hat{a}) + D(\hat{b})$.
> > > >
> > > > **MCTN**:  Given $(a, b)$, model $a  \rightarrow \hat{b}  \rightarrow \hat{a}$, and calculate $L(a, \hat{a}) + L(b, \hat{b})$.
> > > >
> > > > **LoReTTa**: Given $(a, b)$, model $b  \rightarrow \hat{c}  \rightarrow \hat{a}$, and calculate $L(a, \hat{a})$.
> > > >
> > > > Thus, we **do not simply add another step** to cycle consistency (as in CycleGAN and MCTN) by extending $a  \rightarrow \hat{b}  \rightarrow \hat{a}$ to $a  \rightarrow \hat{b} \rightarrow \hat{c} \rightarrow \hat{a}$. In fact, we model $b  \rightarrow \hat{c}  \rightarrow \hat{a}$. Intuitively, this ensures that the model does not "cheat" by memorizing the input $a$ in order to reconstruct $a$. We also **do not use a cycle loss** since $a$ is not used as an input and output. Thus, our loss $L(a, \hat{a})$ is not cyclic with respect to the model's inputs. However, we still make sure that the predicted modality is consistent with the data as a whole. This has a similar flavor to cycle consistency, but as can be seen above, it is not the same.
> > > >
> > > > We would now like to compare LoReTTa with MCTN by Pham et al. (2019), which we are happy to include in our manuscript. The paper proposes to learn a joint multimodal representation of multiple modalities. For this purpose, the authors assume that there is one source modality and multiple target modalities. An encoder RNN is trained to encode the source modality $S0$ into a common embedding space, a decoder RNN is then tasked to predict the first target modality $T1$. The target modality T1 is used analogously to predict the source modality $S0$ – achieving cycle consistency. For the second target modality $T2$ the joint embedding of $S0$ and $T1$ is fed to another encoder RNN to obtain the joint embedding of $S0$, $T1$, and $T2$, which is used by yet another decoder RNN to predict the second target modality $T2$. However, no cycle consistency is applied to $S0$ and $T2$.
> > > >
> > > > We will now highlight the similarities between this work and our own: Both approaches assume a linking modality and use cross-modal translation. In particular, MCTN use the traditional idea of cycle consistency by translating between two domains/modalities. Therefore, they have two encoder-decoder RNNs and two joint embeddings: one for $S0 \leftrightarrow T1$ and one for $(S0 \leftrightarrow T1) \rightarrow T2$. This final representation, used for downstream tasks, can be obtained by using $S0$ as input – but not $T1, T2, (S0, T1), (S0, T2), (T1, T2)$, or $(S0, T1, T2)$, as shown in the experiments section. Thus, while their method effectively learns a multimodal embedding space, it cannot exploit different combinations of modalities at inference time. As shown in our experiments, LoReTTa allows for such an approach. Moreover, the joint multimodal space is not transitive $S0 \leftrightarrow T1 \leftrightarrow T2$, there is no interaction between all modalities, and the whole model requires an additional encoder-decoder for each new modality. **Most importantly**, MCTN must be trained with all three modalities present. This is a strong availability assumption. As shown in our experiments, LoReTTa does not require three fully aligned modalities. One linking modality is sufficient to train our model.

---

> > > > > ### Author Response · Authors · 2023-08-14
> > > > >
> > > > > **References**
> > > > >
> > > > > Artetxe et al., On the Role of Bidirectionality in Language Model Pre-Training, 2022
> > > > >
> > > > > Pham et al., Found in translation: learning robust joint representations by cyclic translations between modalities, 2019
> > > > >
> > > > > Raffel et al., Exploring the Limits of Transfer Learning with a Unified Text-to-Text Transformer, 2019
> > > > >
> > > > > Wang and Chuo, BERT has a Mouth, and It Must Speak: BERT as a Markov Random Field Language Model, 2019
> > > > >
> > > > > Wang et al., SimVLM: Simple Visual Language Model Pretraining with Weak Supervision, 2021

---

> > > > > ### Comment · Reviewer_QEx4 · 2023-08-15
> > > > >
> > > > > __(2.0)__ I thank the authors for the additional discussion, which I would highly recommend to add to their manuscript. Please don't get me wrong, I believe the proposed approach constitutes an interesting extension of cycle consistency and is a valuable contribution. By more explicitly contrasting it to prior work as done in the answer by the authors above, the benefits become much clearer and it will be easier for readers to understand the similarities and the crucial differences, and such a discussion thus significantly improves the paper.
> > > > >
> > > > > Given its similarities, a discussion of the MCTN approach cannot be missing and should definitely be added to the final version by the authors.

---

> > > > > > ### Author Response · Authors · 2023-08-16
> > > > > >
> > > > > > We thank the reviewer for the very productive discussion. In pursuit of a holistic understanding and driven by scientific curiosity, we have performed the suggested experiments. Specifically, we applied transitive modeling directly to GPT without using commutative modeling or causal masking to obtain T-GPT. Furthermore, we iteratively use the MASK tokens of BERT to generate the missing modalities required for a transitive version of BERT (T-BERT). The results below clearly show that both models benefit from transitive modeling in most cases. However, consistent with the discussion above and the literature, BERT fails to generate long-range and high-quality samples, as seen in the results with mRNA (seq_len = 2048). We will be happy to include the new findings and insights in the final manuscript. This includes the additional experiments, a comparison between cycle consistency and transitive modeling, and a discussion about MCTN.
> > > > > >
> > > > > > Method | mRNA | miRNA | RPPA | mRNA-miRNA | mRNA-RPPA | miRNA-RPPA | mRNA-miRNA-RPPA
> > > > > > |----------|----------|----------|----------|----------|----------|----------|----------|
> > > > > > BERT | 0.592 | 0.621 | 0.594 |  $\phantom{....}$ 0.595 | $\phantom{....}$ 0.613 | $\phantom{....}$ 0.594 | $\phantom{........}$ 0.610 |
> > > > > > T-BERT | 0.573 | 0.618 | $\textbf{0.626}$ |  $\phantom{....}$ 0.580 | $\phantom{....}$ 0.591 | $\phantom{....}$ $\textbf{0.623}$ | $\phantom{........}$ 0.608 |
> > > > > > GPT | 0.575 | 0.590 | 0.611 |  $\phantom{....}$ 0.585 | $\phantom{....}$ 0.576 | $\phantom{....}$ 0.606 | $\phantom{........}$ 0.588 |
> > > > > > T-GPT | 0.616 | 0.607 | 0.599 |  $\phantom{....}$ 0.620 | $\phantom{....}$ 0.619 | $\phantom{....}$ 0.611 | $\phantom{........}$ 0.622 |
> > > > > > CLIP | 0.561 | 0.603 | 0.587 |  $\phantom{....}$ 0.610 | $\phantom{....}$ 0.600 | $\phantom{....}$ $\textbf{0.623}$ | $\phantom{........}$ 0.612 |
> > > > > > C2M3 | 0.621 | 0.599 | 0.599 |  $\phantom{....}$ 0.624 | $\phantom{....}$ 0.620 | $\phantom{....}$ 0.571 | $\phantom{........}$ 0.624 |
> > > > > > LoReTTa | $\textbf{0.652}$ | $\textbf{0.623}$ | 0.563 |  $\phantom{....}$ $\textbf{0.660}$ | $\phantom{....}$ $\textbf{0.652}$ | $\phantom{....}$ $\textbf{0.623}$ | $\phantom{........}$ $\textbf{0.657}$ |

---

> > > > > > > ### Comment · Reviewer_QEx4 · 2023-08-16
> > > > > > >
> > > > > > > I am very grateful that the authors took the time to run these additional experiments during the very limited time of the rebuttal period. I believe these results significantly strengthen the authors' contributions and resolve my concerns W1 and W2.
> > > > > > >  Therefore, and in light of the other additional results provided by the authors during the rebuttal period, I increase my score to 'Weak Accept', as I think the authors make a valuable contribution.
> > > > > > >
> > > > > > > My main hesitation for increasing my score even further stems from W3 and W4, i.e., from the accessibility of the paper to fellow researchers; this concern is of course difficult to resolve within the rebuttal period. If, additionally to the new results provided in the rebuttal, the authors include the additional discussion provided [in this answer](https://openreview.net/forum?id=nArzDm353Y&noteId=1Of69PsfuL) to better contextualise their approach with respect to related work, as well as ensure that all training details for reproducing their results are adequately and accessibly described, I believe the submission is strong enough to be accepted at NeurIPS.
> > > > > > >
> > > > > > > I thank the authors for the fruitful discussion and wish them good luck with their submission.

---

> > > > ### Comment · Reviewer_QEx4 · 2023-08-15
> > > >
> > > > __(1.1)__ I thank the authors for the clarification. Having established that BERT can also be used as a generative model in the proposed setting (in contrast to what the authors wrote in the first rebuttal 'By design of transitive modeling, the missing modality must be generated. This is not possible with BERT.'), I would like to come back to my original comment on the submission in __W1__: 'It seems to me like the submission could be strengthened by showing that all of the baselines benefit from the addition of cycle consistency [i.e. transitive modelling]'. Specifically, as the transitive modelling is the main novelty of the submission, highlighting its general applicability for improving the learnt representations of multiple baselines would strengthen the paper.
> > > >
> > > > __(1.2)__ Looking at Table 3, BERT yields results that are very competitive to those of C2M3 and GPT. As such, I find the statement 'Since the goal of transitive modeling is to obtain high quality predictions of the missing modality, a causal approach is the best approach, as evidenced in the literature.' not fully convincing. Again, as I stated in my original review, I believe the submission would be strengthened if this claim were to be explicitly evaluated. If it holds true, the benefits or LoReTTa would be more convincing — if not, this would further show the power of transitive modelling, which is the main contribution, and therefore also strengthen the paper.

---

### Official Review · Reviewer_u4o4 · 2023-07-15

**Soundness:** 1 poor
**Presentation:** 2 fair
**Contribution:** 2 fair
**Rating:** 5
**Confidence:** 3

**Summary:**

The paper proposes a new self-supervised learning method to pretrain from multiple modalities. More specifically, the paper argues that it is hard to obtain datasets with 3 aligned modalities (e.g., text, images, and speech) and tries to address this issue. This can be especially cumbersome for medical datasets where some modalities can be missing for certain patients. As a result, this work proposes a self-supervised framework based on (causal) masked modeling, commutativity, and transitivity to go from one modality to another. These can be regarded as additional consistencies that are imposed during the pretraining stage. While the pretraining only has seen disjoint modality pairs, the model is able to handle any modality combination at test time. The evaluation considers the perplexity metric and also the classification score for unseen modality pairs during the pretraining stage.

**Strengths:**

- Focus: The premise and focus of the paper is valuable as it addresses a real-world problem in order to pretrain on multiple modalities. The paper starts with the argument that it is  hard to collect perfectly aligned datasets when aiming to pretrain on multiple modalities. This alignment step is indeed a very difficult procedure that can cost a significant amount of time and money. Furthemore, in some cases, aligning the data might not even be possible due to missing information/modalities, such as in medical datasets. For instance, the paper considers mRNA, miRNA, and RPPA samples from the TCGA dataset (a medical dataset) when a certain modality is missing. This could be useful in practice.


- Method: To address the aforementioned issue, the approach relies on causal masked modeling and cycle consistency, which are powerful tools that I have not seen being combined.
- Style: The paper is well-written and easy to follow. The overview figure (see Figure 2)  is especially valuable to quickly grasp the aim and mechanics of the presented approach.


- Experiments: The experiments show that the approach can improve over strong baselines, such as BERT and GPT (see Table 3) and can leverage multiple modalities when some might be missing. Advantageously the presented method can leverage multiple modalities as these simply need to be tokenized, such as audio, images, and speech.

**Weaknesses:**

- The scale and scope of the experiments is limited: As the paper considers pretraining, the paper only considers two datasets with a fairly small amount of samples. For instance the SVL-MNIST and TCGA-OMICS datasets are small-scale datasets with respectively 12k and 3.5k sample pairs. Furthermore, only limited finetuning experiments  are considered. What about other tasks, such as retrieval or (visual) question answering? Competing works such as CLIP and BERT have shown to work well for these tasks as well. Overall, it is difficult to judge the generality of the approach as the scope of the experiments is limited.

- The originality of the approach is fairly limited as the approach can be seen as a combination of ideas in prior works. For example, the causal masking strategy is similar to [1, 28, 11], and the transitive loss is similar to cycle consistency [90]. While the paper mentions modification to these works in the paper (L141 and L153), it is not clear how significant these contributions are to the final performance. More information would be useful.

- It can be observed from Table 3 that CLIP already obtains relatively good results compared to the proposed method. More information on why this is the case would be useful. While CLIP requires the pretraining of 3 different encoders, as the paper mentions, I don’t think this is a strong argument. Especially, as the paper under review also relies on expensive ways to compress high-dimensional data with a VQVAE (see L116). More information on the computational cost  when comparing the approach to CLIP would be valuable during the rebuttal.

- There is currently no experiment that includes all modalities during pretraining. This could be seen as an upperbound but is currently missing as mentioned in the limitations.

**Questions:**

Please address the questions and issues raised in the Weaknesses section.

**Limitations:**

The paper briefly discusses the limitations, environmental impact and broader impact at the end of the paper.

---

> ### Author Rebuttal · Authors · 2023-08-07
>
> **Dear Reviewer u4o4,**
>
> Thank you for pointing out that our proposed method improves on strong baselines such as BERT and GPT by combining powerful ideas and applying them to real-world problems. In the comments below, we address each of the remaining concerns that were raised.
>
> **Scale and scope of the experiments.** We chose SVL-MNIST to perform extensive experiments with a limited computational budget. In total, we have trained 12 models and refined them 162 times on this dataset alone. This does not even include the search for hyperparameters. We complemented this with experiments on a real medical dataset for the important task of survival prediction. In both cases, LoReTTa works as theorized in Section 4. To assure the Reviewer that our method also extends to larger datasets, we performed an additional experiment (cross-modal translation) on the MUGEN [Hayes et al., 2022] dataset with 375,000 aligned video, audio, and text samples. We refer to the global rebuttal for the results, which show that LoReTTa is generalizable and scales very well to larger datasets and other tasks.
>
> **Originality of the approach.** We noted in the paper that causal modeling and masked modeling are two of the current state-of-the-art methods for self-supervised learning. Both methods had been combined into a strategy called causal masked modeling [Aghajanyan et al., 2022; Bavarian et al., 2022; Fried et al., 2023], which we used as a starting point and never claimed as our invention. However, we greatly improved causal masked modeling by extending it with commutative modeling (C2M3) and transitive modeling (LoReTTa), which had never been done before. By comparing LoReTTa with C2M3, GPT [Radford et al., 2018], and BERT [Devlin et al., 2018], we indirectly included an ablation study. In fact, transitive modeling is a very unique contribution. On the surface, it shares similarities with cycle consistency [Zhu et al., 2017]. But despite this high-level conceptual relationship, the two approaches work quite differently. For example, CycleGAN is only able to transition between two domains (image styles) of the same modality (images), while LoReTTa works on three or more modalities, e.g. image, video, audio, and text, with complex multimodal relations between them. This also helps to avoid representation collapse, which is common in adversarial training. We have given a more detailed explanation in the global rebuttal.
>
> **About CLIP results.** LoReTTa consistently outperforms CLIP [Radford et al., 2022]  in 5 out of 7 modality combinations (see paper) by a large c-index. Only in one case (R), CLIP outperforms LoReTTa and in another (I, R) it is equal. In both cases, the reverse-phase protein array (R) is included as a modality. For GPT, BERT, C2M3, and LoReTTa, the combination (I, R) leads to a worse prediction than I or R alone. Only for CLIP the c-index increases. We assume that I and R lead to negative transfer, but since CLIP learns to create similar embeddings for both, the embedding vectors may be devoid of features causing the problem. This could be considered a kind of smoothing or regularization. However, this could also cause the model to ignore important high-frequency features inherent in each modality, leading to worse performance as seen for the other modality combinations. Another issue with self-supervised contrastive learning is that two samples with the same label might be pushed away in the embedding space. For example, a batch might contain an image of a dog with a corresponding caption and another image of a similar dog with another similar caption. The CLIP loss only learns to pull the feature vector of each pair closer but pushes each sample away. Thus, contrastive learning requires very large batch sizes and huge amounts of training data [Chet et al., 2020]. In addition, each modality requires its own encoder. LoReTTa, however, already works on smaller datasets which is a huge advantage for data-deprived domains like medicine. In addition, it can reuse existing and pre-trained tokenizers.
>
> **Upper bound.** We did not include experiments for a model trained with all three modalities. This was done to emphasize the real-world problem that often there is simply no dataset with all three modalities at once. The upper bound is the current state-of-the-art for existing datasets, which LoReTTa manages to outperform. We have focused our resources on providing extensive experiments for this setting. In the global rebuttal, we have also included an even larger experiment on a new dataset [Hayes et al., 2022] to show the usefulness of LoReTTa on even more problems.
>
> **New results.** We have added the additional large-scale experiment in the global rebuttal. Both GPT and LoReTTa were trained on disjoint (video, audio) and (video, text) pairs to solve the problem of cross-modal translation. We then evaluated the models on the unseen task of audio captioning. As can be seen above, LoReTTa is more than capable of overcoming the modality gap and rivals the upper-bound models (MMGPT).
>
>
> **References**
>
> Aghajanyan et al., CM3: A Causal Masked Multimodal Model of the Internet, 2022
>
> Bavarian et al., Efficient Training of Language Models to Fill in the Middle, 2022
>
> Chen et al., A Simple Framework for Contrastive Learning of Visual Representations, 2020
>
> Devlin et al., BERT: Pre-training of Deep Bidirectional Transformers for Language Understanding, 2018
>
> Fried et al, InCoder: A Generative Model for Code Infilling and Synthesis, 2023
>
> Hayes et al., MUGEN: A Playground for Video-Audio-Text Multimodal Understanding and GENeration, 2022
>
> Radford et al., Improving Language Understanding by Generative Pre-Training, 2018
>
> Radford et al., Learning Transferable Visual Models From Natural Language Supervision, 2022
>
> Zhu et al., Unpaired Image-to-Image Translation using Cycle-Consistent Adversarial Networks, 2022

---

> > ### Comment · Reviewer_u4o4 · 2023-08-12
> > **Questions After Rebuttal**
> >
> > I thank the authors for providing the rebuttal. I still have remaining questions after reading the rebuttal and other reviews:
> >
> > 1. Why are no upper-bound experiments included? While the availability of three modalities might be less likely in practice, it’s still important to include this information in the paper. This upper-bound is currently missing.
> >
> > 2. When and how does the paper rely on expensive ways to compress high-dimensional data, such as with a VQVAE? (see L116). It’s valuable to include more information on the computational cost when comparing the approach to CLIP. Especially, since L321 emphasizes that CLIP requires the pretraining of 3 encoders, one for each modality.
> >
> > 3. Is it possible to evaluate the current approach for tasks like retrieval and VQA problems, as shown in CLIP?

---

> > > ### Author Response · Authors · 2023-08-16
> > >
> > > We thank the reviewer for helping us make our paper as clear, concise, and comprehensive as possible. Below, we would like to answer the three open questions.
> > >
> > > **(1)** We stated in the rebuttal that we focused all of our time and resources on providing the large-scale experiments. In particular, we read the dataset authors' paper, learned the intricacies of processing the dataset, downloaded the dataset, trained the models, and evaluated them on the test dataset. We agree with all the reviewers that this experiment is important, and we provided it. Upon completion, we immediately started the “upper bound experiments”. Both training and testing have been completed and are shown below. We trained this model using C2M3 on all fully-aligned modalities. We refer to it as “C2M3 (3-modal)” and not “Upper Bound Model” since training with all modalities does not guarantee the highest possible result due to negative transfer and modality competition, which can be mitigated by more advanced techniques such as transitive modeling as shown by LoReTTa. We have also added transitive modeling to BERT and GPT, called T-BERT and T-GPT, to better demonstrate the significance of our contribution.
> > >
> > > **(2)** We briefly mentioned the tokenization strategy in the Methods and Experiments section. In the final paper, we will be more specific to avoid confusion. For the SVL-MNIST experiments, we considered each byte stream as a token and binned them to integer values. For the high-dimensional TCGA-OMICS dataset, we first reduced the input dimension via PCA and also discretized the values. For the large-scale MUGEN dataset, we used the pre-trained VQ-VAE encoders for images and audio provided along the dataset to obtain the discrete representations.
> > >
> > > **(3)** We have analyzed our proposed method on three very different datasets, analyzing different metrics such as perplexity, accuracy, c-index, BLEU4, METEOR, and ROUGE. Despite this large diversity of tasks, we have consistently shown that LoReTTa outperforms current methods and comes close to the theoretical upper bound. Although not exactly equivalent, in the MUGEN experiments we have applied LoReTTa to the task of cross-modal translation from audio to text which is very similar to audio captioning. While this is not exactly visual or acoustic question answering, it serves as a good proxy since we are not aware of a large-scale visual-acoustic question answering dataset with three modalities. Evaluating our method in an exhaustive fashion on even more tasks would be out of the scope of this work.
> > >
> > > Method | mRNA | miRNA | RPPA | mRNA-miRNA | mRNA-RPPA | miRNA-RPPA | mRNA-miRNA-RPPA
> > > |----------|----------|----------|----------|----------|----------|----------|----------|
> > > BERT | 0.592 | 0.621 | 0.594 |  $\phantom{....}$ 0.595 | $\phantom{....}$ 0.613 | $\phantom{....}$ 0.594 | $\phantom{........}$ 0.610 |
> > > T-BERT | 0.573 | 0.618 | $\textbf{0.626}$ |  $\phantom{....}$ 0.580 | $\phantom{....}$ 0.591 | $\phantom{....}$ $\textbf{0.623}$ | $\phantom{........}$ 0.608 |
> > > GPT | 0.575 | 0.590 | 0.611 |  $\phantom{....}$ 0.585 | $\phantom{....}$ 0.576 | $\phantom{....}$ 0.606 | $\phantom{........}$ 0.588 |
> > > T-GPT | 0.616 | 0.607 | 0.599 |  $\phantom{....}$ 0.620 | $\phantom{....}$ 0.619 | $\phantom{....}$ 0.611 | $\phantom{........}$ 0.622 |
> > > CLIP | 0.561 | 0.603 | 0.587 |  $\phantom{....}$ 0.610 | $\phantom{....}$ 0.600 | $\phantom{....}$ $\textbf{0.623}$ | $\phantom{........}$ 0.612 |
> > > C2M3 | 0.621 | 0.599 | 0.599 |  $\phantom{....}$ 0.624 | $\phantom{....}$ 0.620 | $\phantom{....}$ 0.571 | $\phantom{........}$ 0.624 |
> > > LoReTTa | $\textbf{0.652}$ | $\textbf{0.623}$ | 0.563 |  $\phantom{....}$ $\textbf{0.660}$ | $\phantom{....}$ $\textbf{0.652}$ | $\phantom{....}$ $\textbf{0.623}$ | $\phantom{........}$ $\textbf{0.657}$ |
> > > | | | | | | | | |
> > > C2M3 (3-modal) | 0.665 | 0.635 | 0.645 | $\phantom{....}$ 0.643 | $\phantom{....}$ 0.671 | $\phantom{....}$ 0.650 | $\phantom{........}$ 0.620 |
> > > | | | | | | | | |
> > >
> > > Method | IMG | TXT | WAV | IMG-TXT | IMG-WAV | TXT-WAV | IMG-TXT-WAV
> > > |----------|----------|----------|----------|----------|----------|----------|----------|
> > > LoReTTa$_I$ | 82.7 | 62.8 | 82.5 | $\phantom{..}$ 88.5 | $\phantom{..}$ 89.7 | $\phantom{..}$ $\textbf{84.0}$ | $\phantom{....}$ $\textbf{90.7}$ |
> > > LoReTTa$_T$ | 81.2 | 63.3 | 81.0 | $\phantom{..}$ 89.0 | $\phantom{..}$ $\textbf{80.9}$ | $\phantom{..}$ 85.5  | $\phantom{....}$ $\textbf{87.8}$ |
> > > LoReTTa$_W$ | 80.1 | 62.1 | 84.2 | $\phantom{..}$ $\textbf{83.0}$ | $\phantom{..}$ 90.4 | $\phantom{..}$ 89.0 | $\phantom{....}$ $\textbf{91.6}$ |
> > > | | | | | | | | |
> > > C2M3 (3-modal)  | 80.8 | 56.6 | 82.4 | $\phantom{..}$ 84.9 | $\phantom{..}$ 86.3 | $\phantom{..}$ 84.9 | $\phantom{....}$ 88.3 |
> > > | | | | | | | | |

---

> > > > ### Comment · Reviewer_u4o4 · 2023-08-21
> > > > **Response**
> > > >
> > > > Thank you for discussion.
> > > >
> > > > However, it's still not clear to me if L116 in the submission is correct if a VQVAE is being used. Please specifiy the computational cost compared to CLIP in further updates. Based on the above discussion and addressed concerns wrt the upperbound, I will raise my score to 5.

---

### Official Review · Reviewer_a22R · 2023-08-01

**Soundness:** 2 fair
**Presentation:** 2 fair
**Contribution:** 2 fair
**Rating:** 4
**Confidence:** 3

**Summary:**

This paper proposes a self-supervised learning framework, LoReTTa (Linking mOdalities with a tRansitive and commutativE pre-Training sTrAtegy), for multimodal learning with not-aligned multimodal datasets. To be specific, LoReTTa (i) first trains an autoregressive model for generating A->B and B->C, and then (ii) generates pseudo targets for A->C using the model. Then, it (iii) performs autoregressive modeling again. This paper demonstrates that this transitive modeling can improve generation performance between the unseen pair of modalities (i.e., A->C).


**Strengths:**

- This paper is well-motivated.
- The proposed framework is widely applicable.
- The proposed framework outperforms baselines that do not consider transitive modeling.


**Weaknesses:**

- Limited methodological novelty
  - The proposed framework is mainly based on the idea of CycleGAN, and it simply uses causal language/mask modeling with Transformers. I agree that the idea is more general, but I feel some lack of novelty. I'm also concerned about the scalability of the framework.
  - In addition, due to the recent advance of foundation models (e.g., Llama2 [1], DINOv2 [2]), there are many research attempts to utilize them for multimodal learning. For example, Chameleon [3] uses expert models for multi-modalities (see [4] for a survey on multimodal LLMs). I think the authors should discuss what is the advantage of the proposed framework over the approach utilizing one foundation model as a general knowledge hub.
- Weak theoretical analysis (Section 4)
  - The analysis is too vague. Any theoretical statement is not stated concretely. Also, the authors should discuss the generalization error bound because, for example, the sample (C0...C3) in Figure 2c is not observed.
  - Notations are somewhat confusing. Why do the authors state the definition (3)? $f(x_1|x_2)=x_1+e$ is a mathematically correct notation? Many notations are roughly written, so it may cause confusion to the readers.
  - Section 4 should be reorganized.
- Weak experiments
  - In this topic, large-scale experiments and diverse applications are important. However, I think the provided experiments are very small-scale, e.g., MNIST images. I suggest using more challenging benchmarks, e.g., [5-6].

[1] Llama 2: Open Foundation and Fine-Tuned Chat Models \
[2] DINOv2: Learning Robust Visual Features without Supervision \
[3] Chameleon: Plug-and-Play Compositional Reasoning with Large Language Models \
[4] A Survey on Multimodal Large Language Models \
[5] Multi-modal Dense Video Captioning, CVPR Workshop 2020 \
[6] MUGEN: A Playground for Video-Audio-Text Multimodal Understanding and GENeration, 2022


**Questions:**

1. About causal masked modeling (L140-L143)
   - If randomly mask some tokens and move them to the end, then such a prediction task is somewhat different from the original masked language modeling (like BERT) since it loses positional information about the masked tokens. Is the approach ok? I also think it may interfere with the causal language modeling.
2. In Equation (3), what is $L_X$?



**Limitations:**

This paper has addressed the limitations and the potential impacts.

---

> ### Author Rebuttal · Authors · 2023-08-07
>
> **Dear Reviewer a22R,**
>
> We appreciate your valuable comments on our paper. Your recognition of the motivation and broad applicability of our proposed framework is very encouraging. Below, we would like to address the open questions.
>
> **Limited methodological novelty.** In the global rebuttal, we pointed out that LoReTTa is much more general and applicable than CycleGAN [Zhu et al., 2017]. We used CycleGAN as an analogy because it has a similar flavor to our method. But beyond the high-level conceptual similarity, our transitive modeling approach is fundamentally different. For example, it can be used to transition across multiple modalities – not just two domains from the same modality. We use a single transformer as the central point for all modalities. By combining causal, masked, commutative, and transitive modeling, we have shown theoretically and experimentally that our model can meaningfully integrate information from different modalities. This is in contrast to recent approaches that use LLMs as a communication hub to explain and reason about one modality. For example, Chameleon [Lu et al., 2023] uses different modalities in a plug-and-play approach to describe images, process spreadsheets, or search the web. However, this approach does not effectively integrate different modalities to potentially gain new insights. In our TCGA-OMICS experiments, we have shown that combining genomic, transcriptomic, or proteomic data provides valuable prognostic results. Adding each modality individually to a foundation model is not enough, as they do not interact with each other. Thus, LoReTTa serves as a good starting point for fine-tuning these general models. One could also think about using LLMs that communicate with LoReTTa via API calls to solve more challenging problems that a user faces with their multimodal input.
>
> **Weak theoretical analysis.** Our theoretical analysis is supported by references and highlights the probabilistic principles behind our approach. It also discusses the error propagation caused by transitive modeling and how we mitigate this by modeling the omitted modality. We use standard notation from stochastic, analysis, and linear algebra to emphasize the intricate details that arise. Any confusion caused by this will be addressed in an updated version of the manuscript. Regarding the unobserved samples $(C0, ..., C3)$, they are indeed missing, but the goal of LoReTTa is precisely to impute this missing modality and align it with the existing data points. It corresponds to the predicted sample $\hat{x}_3 = x_3 + e$ in Equation 6 for which we have provided the error bounds.
>
> **Weak experiments.** We agree that large-scale experiments and diverse applications are very important. That is why we have provided results on a real medical dataset for the crucial task of survival prediction. Our results show that the proposed approach works not only on synthetic data but also on complex biomedical samples. We are very grateful to the Reviewer for suggesting the MUGEN [Hayes et al., 2022] dataset. It is indeed a challenging and large-scale benchmark. After hearing about it, we immediately downloaded the dataset and started training. We summarized the results in the global rebuttal, which shows that LoReTTa works on even more problems and is easily scalable. These results will be included in the final version of the manuscript.
>
> **New results**. The table below shows the results of our new experiments. Both GPT and LoReTTa were trained on disjoint (video, audio) and (video, text) pairs to solve the problem of cross-modal translation. We then evaluated the models on the unseen task of audio captioning. As can be seen, LoReTTa is more than capable of overcoming the modality gap and rivals the upper-bound models (MMGPT). More details can be found in the global rebuttal.
>
> | Method $\phantom{.}$ | $\phantom{...}$ Train | $\phantom{a}$ Test | BLEU4 | METEOR | ROUGE |
> |----------|----------|----------|----------|----------|----------|
> | GPT | A $\rightarrow$ V, V $\rightarrow$ T | $\phantom{.}$ A $\rightarrow$ T |  $\phantom{...}$ 1.7 |  $\phantom{...}$ 18.5 | $\phantom{...}$  30.7 |
> | LoReTTa | $\phantom{.}$ A $\leftrightarrow$ V $\leftrightarrow$ T | $\phantom{.}$ A $\rightarrow$ T | $\phantom{...}$ 2.8 |  $\phantom{...}$ 20.8 |  $\phantom{...}$ 34.7 |
> |     |     |     |     |
> | MMGPT | $\phantom{...}$ A $\rightarrow$ T | $\phantom{.}$ A $\rightarrow$ T | $\phantom{...}$ 6.7 | $\phantom{...}$ 19.4 | $\phantom{...}$ 27.1 |
> | MMGPT| $\phantom{...}$ V $\rightarrow$ T |$\phantom{.}$  V $\rightarrow$ T | $\phantom{...}$ 7.8 | $\phantom{...}$  21.3 | $\phantom{...}$ 29.1 |
>
>
> **References**
>
> Hayes et al., MUGEN: A Playground for Video-Audio-Text Multimodal Understanding and GENeration, 2022
>
> Lu et al., Chameleon: Plug-and-Play Compositional Reasoning with Large Language Models, 2023
>
> Zhu et al., Unpaired Image-to-Image Translation using Cycle-Consistent Adversarial Networks, 2022

---

### Author Rebuttal · Authors · 2023-08-07

**Dear Area Chair and Reviewers,**

We are very grateful for your insightful comments and sincerely appreciate the time and effort you have taken to provide constructive feedback. It is gratifying to hear that all reviewers unanimously agree that the paper is well-written and well-motivated. They acknowledge that our proposed method is "widely applicable" [a22R] and "valuable as it addresses a real-world problem" [u4o4]. In particular, our "results show convincing and significant improvements over the chosen baseline models" [QEx4] and “would be a good baseline for any future methods" [7Goz]. We are also thrilled to highlight that the "theoretical analysis contributes to the understanding" [FSDR] of our novel multi-modal and self-supervised framework LoReTTa. Moreover, the extensive experiments "show that the proposed system is very effective" and supported by "careful ablations against mainstream popular pre-training objectives” [SR2A]. Below, we would like to respond to the most common concerns raised by reviewers. We have also addressed individual questions in the comments section.

**Extending Transitive Modeling to BERT and GPT.**
We would like to clarify the differences between LoReTTa, C2M3, GPT, and BERT, as this has caused some confusion. BERT uses masked modeling, which has proven to be a powerful approach for training encoders and bidirectional representations. GPT, on the other hand, uses causal modeling and unidirectional representations to train autoregressive decoders. GPT excels at generative tasks but has more difficulty with discriminative tasks as compared to BERT. Conversely, BERT is not designed to solve generative problems which is also a drawback of the contrastive method CLIP. FIM and InCoder combine the strengths of both BERT and GPT by introducing causal masked modeling, which reinterprets masked modeling as a causal problem. Thus, we get a GPT-style architecture augmented by BERT-style training. This method was extended to the multimodal case by CM3. We go further and apply commutative modeling to CM3, yielding C2M3. The next step is to incorporate transitive modeling and we arrive at LoReTTa. Thus, LoReTTa unifies BERT and GPT training by combining and extending them. In essence, C2M3 is an extended version of GPT, while BERT and CLIP cannot be directly modified with transitive modeling because they are unable to generate the missing modality. For example, one would have to train a diffusion model on top of the CLIP embeddings to generate an image as in DALLE-2.

**Comparison to CycleGAN.**
We agree that LoReTTa has a similar flavor to the cycle consistency concept introduced in CycleGAN. However, our proposed approach is much more general and applicable. The main idea of CycleGAN is to translate an image from domain A to domain B, compute the adversarial loss on the generated image, and reconstruct the original image from the generated image. This involves two domains from the same modality. On the other hand, LoReTTa works on three or more domains from three or more modalities. In addition, we avoid modeling a → b'→ a as in CycleGAN. Instead, we use the idea of transitivity to model a → b' → c, given the pair (a, c). This is a stronger learning signal since we have to reconstruct an entirely new data point c that was not part of the model input (a). In fact, this idea can be generalized to more complex multimodal relationships such as (A, B), (A, D), (A, C), and (B, C) (as shown in Figure 2d). As long as there is a path from one modality to the other, we can use the principle of transitivity to approximate the full data distribution. How would we learn the missing distribution (C, D)? By taking a real data point (a, d) and learning the consistent transition a → b' → c' → d or a → c' → d, respectively. This would not be possible at all using only the CycleGAN approach.

**Large-scale Experiments.**
We appreciate the reviewer's insights and understand the potential value of large-scale experiments. However, due to resource limitations, we were faced with constraints on the amount of computations. Nevertheless, not counting hyperparameter search, we pre-trained 17 models and fine-tuned them a total of 169 times to obtain a comprehensive experimental insight into the operation of LoReTTa that strongly complements our theoretical analysis. Within these constraints, we were able to derive meaningful and impactful results that contribute significantly to the ongoing scientific conversation in our rapidly changing field. We are fully committed to our research and recognize the importance of scalability and generalizability. Reviewer a22R recommended the new large-scale and multimodal MUGEN dataset with 375K naturally aligned video (V), audio (A), and text (T) samples from a real reinforcement learning problem.

We immediately downloaded all the files and started to run an additional set of experiments. In particular, we used video as the linking modality and considered the disjoint datasets (A, V) and (V, T). We pre-trained the same transformer as the state-of-the-art upper baseline MMGPT. As can be seen below, LoReTTa clearly manages to rival the upper bound.

| Method $\phantom{.}$ | $\phantom{...}$ Train | $\phantom{a}$ Test | BLEU4 | METEOR | ROUGE |
|----------|----------|----------|----------|----------|----------|
| GPT | A $\rightarrow$ V, V $\rightarrow$ T | $\phantom{.}$ A $\rightarrow$ T |  $\phantom{...}$ 1.7 |  $\phantom{...}$ 18.5 | $\phantom{...}$  30.7 |
| LoReTTa | $\phantom{.}$ A $\leftrightarrow$ V $\leftrightarrow$ T | $\phantom{.}$ A $\rightarrow$ T | $\phantom{...}$ 2.8 |  $\phantom{...}$ 20.8 |  $\phantom{...}$ 34.7 |
|     |     |     |     |
| MMGPT | $\phantom{...}$ A $\rightarrow$ T | $\phantom{.}$ A $\rightarrow$ T | $\phantom{...}$ 6.7 | $\phantom{...}$ 19.4 | $\phantom{...}$ 27.1 |
| MMGPT| $\phantom{...}$ V $\rightarrow$ T |$\phantom{.}$  V $\rightarrow$ T | $\phantom{...}$ 7.8 | $\phantom{...}$  21.3 | $\phantom{...}$ 29.1 |

---

### Decision · Program_Chairs · 2023-09-21

**Decision:**

Accept (poster)

**Comment:**

This work proposes a multi-modal learning framework for non-aligned modalities, linking modalities A and C via A-B and B-C. The reviewers enjoyed the paper's performance, simplicity and importance and clear presentation. While some questions about additional datasets, upper-bounds, messaging (about BERT's ability to generate text), these were sufficiently cleared in the extensive rebuttal and discussions.